



# Influence of Oxygen Minimum Zone on Macrobenthic Community Structure in the Northern Benguela Upwelling System: A Macro-Nematode Perspective

Hashim Said Mohamed[1], Beth Wangui Waweru[2], Agnes Muthumbi[3]

[1]Tokyo University of Marine Science and Technology, 5-7, Konan-4, Minato, Tokyo

108-8477

[2]Ghent University, Department Biotechnology, Proeftuinstraat 86, B-9000 Gent, Belgium

[3]Nairobi University, School of Biological and Physical Sciences, Department of Biological Sciences, PO Box 30197-30100, Nairobi, Kenya

Corresponding author: sayeed.said01@yahoo.com





## Abstract

Macrobenthic samples were collected offshore Namibia on board *R/V Mirabilis* during the 3rd
RGNO training and the National Marine Information and Research Centre's (NatMIRC's) plankton
survey from 13th May to 17th May 2016. Two transects, Cape Frio (20º S) and Walvis Bay (23º S),
hosted three stations each, while the third transect, Luderitz (26º S), hosted only one station. From
the results, three oxygen zones were identified, namely Microxic ($<0.1$ ml l$^{-1}$), Dysoxic (0.1-1.0
ml l$^{-1}$), and Oxic ($>1.0$ ml l$^{-1}$). A total of 20 Macrobenthic taxa were identified; Nematoda,
Polychaeta, Cumacea, and Oligochaeta were the most dominant taxa and recorded abundances in
all oxygen zones. Eighteen genera of macro-Nematoda were identified; *Desmolaimus* and
*Paracomesoma* dominated in all oxygen zones, *Metoncholaimus* recorded higher abundance in the
Dysoxic stations, and no abundance in the Oxic stations, and the opposite was observed for
*Halanonchus* and *Dorylaimopsis*. H′ Diversity values for both the general Macrofauna and Macro-
Nematoda were higher in the Oxic stations and lower in both the Dysoxic and Microxic stations,
while an opposite trend was observed for Dominance values. Density values were lower in
Microxic stations and higher in Dysoxic stations, while at the Oxic stations, the density values fell
in between the two hypoxic zones.
**Keywords**: Macrofauna, Macro-Nematoda, Oxygen minimum zone, Benguela Upwelling System
(BUS), Dissolved oxygen, hypoxia





## 1.0 Introduction

Hypoxia is ranked among the major threats to the actualization of the blue economy and the
achievement of the 14th pillar of the Sustainable Development Goals (SDGs), specifically target
14.7 which aims at increasing the economic benefits of marine resource utilization through its
sustainable use by developing countries. In recent decades, the concentration of dissolved oxygen
(DO) in the ocean, specifically in the tropics, has been decreasing. This not only increases the size
of areas under hypoxia but also their prevalence (Breitburg et al., 2018). Although hypoxia can
result from natural phenomena like upwelling and thermal stratification, the current expansion of
hypoxic areas is mainly a result of accelerated nutrient inputs in coastal areas which increases algal
production subsequently resulting in higher organic matter production which in turn results in
increased aerobic microbial decomposition lowering the levels of DO in the water (Gobler &
Baumann, 2016). It has been projected that such changes may affect different organisms differently
depending on their tolerance and reactions to lower dissolved oxygen in their habitats (Brodie
Rudolph et al., 2020). Studies on ecosystems with hypoxia as a natural phenomenon can assist in
predicting and understanding how human-induced hypoxia might affect and shape marine
ecosystems in the face of the expansion of marine areas under hypoxia.
The Benguela upwelling system (BUS) is one of the most productive regions of the world's ocean
(Magalhães, 2018). The high productivity provides a huge source of carbon resulting from
photosynthesis, which gradually sinks through the water column resulting in a rapid oxygen loss
due to biochemical oxygen consumption. This consequently causes low dissolved oxygen



concentrations and thus creates a permanent extensive shallow oxygen minimum zone (Bohata &
Koppelmann, 2013; Emeis et al., 2018). When the oxygen minimum zone (OMZ) comes into
contact with the seafloor, it creates a strong oxygen gradient at the benthic zone at depths ranging
between 50 meters and 300 meters, resulting in a hypoxic (<0.5 ml l$^{-1}$) inner shelf (Gibson &
Atkinson, 2003). The oxygen gradient created at the OMZ's benthic zone is believed to primarily
regulate the benthic community distribution and diversity patterns (Zettler et al., 2013; Teuber et
al., 2013).
The term' benthos' refers to organisms living on and in the sediments of the seafloor and are
distinguished based on their sizes as either megafauna, macrofauna, meiofauna, or microfauna
with some taxa appearing in more than one size category. Macrobenthos are part of the benthos
consisting of organisms retained in a 0.5mm sieve but pass through a 2.00mm sieve (Bachelet,

61  1990).

The general trend observed in most OMZs in global oceans (off Walvis Bay, California, and Oman
margin) indicates that the densities of macrofauna generally display a negative response to
reducing oxygen levels within the OMZ with a 30% to 70% reduction in densities in regions with
less than 0.15 ml l$^{-1}$ (Gibson & Atkinson, 2003). Similarly, diversity reduces as oxygen levels
reduce within the OMZ because of the loss of intolerant species and increased dominance of the
tolerant species. Nematodes and some families from the annelid worms have been observed to be
able to tolerate low oxygen, with Nematoda (in the meiofauna group) having 95-99% abundance.
Some polychaetes families like Spinoid, Dorvilleid, and Lumbrinerid can also tolerate low oxygen
in the OMZ by having a high gill surface area for increasing oxygen uptake. On the other hand,
harpacticoid copepods are the most sensitive taxon to hypoxia (Levin et al., 2009; Zeppilli et al.,

72  2015).



It has been observed that the macrofauna diversity is lowest off Walvis Bay, attributed to the
perennial intense OMZ over the shelf, while the diversity northwards off the Kunene River
increases significantly, which was proposed to be a result of moving away from the intense OMZ
cells off Walvis Bay and also the reflection of the biogeography where diversity increases with
reducing latitude. Outside the OMZ, bathymetry, and latitude are said to be the factors affecting
the infaunal communities at the Namibian shelf (Steffani et al., 2015).
This study aims to identify the relationship between the levels of dissolved oxygen and the
macrobenthic community distribution across the Northern Benguela Upwelling system as a guide
on how the expansion of man-made hypoxia might influence the benthic fauna distribution on the
seafloor.
**2.0 Material and Methods**
**2.1 Study Area**
The study area was located across the Northern Benguela Upwelling System (between 26º S and
20º S) along the Namibian continental shelf, which hosts a deep continental shelf (around 300 m).
The intense upwelling in the study site has rendered the area highly productive, resulting in a
characteristic shallow OMZ (50-200 m) with stronger productive cells around Luderitz and Walvis
Bay (Bohata & Koppelmann, 2013). The inner shelf is described to be extremely oxygen-depleted,
caused by in situ organic matter decomposition and warm poleward Angola current, which peaks
in June-July while the continental slope below the OMZ is better oxygenated (Levin et al., 2009;
Emeis et al., 2004).



The benthic zone in the OMZ in Northern BUS is characterized by extensive areas of diatomaceous
mud, which are associated with high surface primary production and low concentration of
dissolved oxygen (Levin et al., 2009; Steffani et al., 2015).
**2.1 Sample Collection**
Samples were collected from three transects (off Luderitz (OL) 26º S, off Walvis Bay (OWB) 23º
S, and off Cape Frio (OCF) 20º S) onboard *R/V Mirabilis* during the RGNO training National
Marine Information and Research Centre's (NatMIRC's) plankton survey from 13[th] May to 17[th]
May 2016. The sampling stations were located at 02 nm, 20 nm, 40 nm, or 70 nm at each transect,
with the 26º S transect hosting only one station at 90 nm. However, benthic samples from these
stations were dependent on the prevailing weather conditions and the ability to get good core
samples (Figure 1).
Sampling was done using a multi-corer from which individual cores were taken and sub-sampled
for macrobenthos using a 6.4 cm diameter corer, and sediment samples for granulometry and
organic matter analysis were taken simultaneously. Replicate samples were taken from a
subsequent deployment of the multi-corer, where possible, to avoid pseudoreplication. The
macrofauna cored samples were put in sampling bottles and immediately fixed with 5% buffered
formalin, while samples for organic matter analysis were frozen to arrest microbial decomposition.
Depth and temperature measurements were collected from a probe attached to the multi-corer,
while dissolved oxygen concentrations in this study used the Winkler method from the overlying
water (Montgomery et al., 1964).



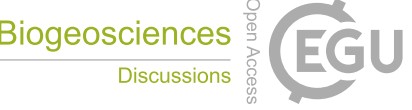

### 2.3 Laboratory analysis


In the laboratory, macrobenthic samples were sieved between 2.00 mm and 0.45 mm sieves, and
what was retained in the 0.45 mm sieve was then preserved in 5% buffered formalin solution, and
3-5 drops of Rose Bengal solution were added to aid in sorting. The samples were then rinsed with
water, sorted under a dissecting microscope using the key from (Thiel & Higgins, 1988), and
counted. The macro-nematodes were then pin-picked later, fixed on permanent slides, and
identified to the genus level using the key from (Platt & Warwick, 1988).
For sediment granulometry, samples were wet-sieved using a 1.00 mm sieve, 0.5 mm, 0.25 mm,
105 mm, 0.063 mm, and those below 0.063 mm (<0.063 mm). The sediments collected at the
different sieves were then put on pre-weighed Petri dishes and dried in an oven at 60º C until a
constant weight was reached, and these dried weights were used for sediment granulometry
(Wentworth, 1922).
The samples for total organic matter (TOM) were dried in an oven at 60º C until a constant weight
was achieved, then a known weight of each sample was then placed in labeled aluminum foil, and
the loss of weight on ignition (LOI) technique was used to determine TOM from the study site
(Thiel & Higgins, 1988).

### 2.4 Data analysis


Out of the seven sampled stations, only three had replicates. Therefore, a data analysis approach
was needed to provide detailed insight into the structure of the macrobenthic communities in the
northern Benguela Upwelling System. The environmental variable that had the highest significant
correlation with the various biotic indices was used to group the stations for proper community
analysis. The data recorded were analyzed for abundance (density and relative abundance) in



Excel, Paleontological Statistics Software package (PAST v2.17c) (Hammer et al., 2001) was used
to calculate the diversity and the diversity t-test between the stations. Bray Curtis similarity,
Analysis of Similarity (ANOSIM), and Similarity percentages (SIMPER) were conducted using
Plymouth Routines in Multivariate Ecological Research (PRIMER v5.2.9) (Clarke & Gorley,

139    2005).

Stepwise regression analysis (using SPSS) was used to determine the environmental variables that
were predictors of the distribution patterns of the macrobenthic diversity patterns in the BUS.
Various diversity indices were calculated using PAST for each station; the various replicates were
considered as a single station, and these indices were then correlated individually with the
environmental variables recorded in the study.
**3.0 Results**
**3.1 Abiotic variables**
Total Organic Matter (%TOM) showed an opposite trend with depth, with higher organic matter
values recorded in the shallower stations where stations 20020, 20002, and 23002 had the highest
TOM of 38.6%±2.16, 27.5%±1.55 and 23.1%, respectively. The lowest TOM values were found
in most offshore stations, stations 23070 and 26090, with 9.89±1.55 and 4.4, respectively (Table
1). Similarly, dissolved oxygen values had lower values recorded in the shallower onshore station.
The lowest oxygen values were recorded on the OWB 23° S transect, with the two most onshore
stations (23020 and 23002) recording the lowest levels of oxygen (0.02 and 0.06 ml l$^{-1,}$
respectively). The inshore station from transect OCF 20$^0$ S (20002) had a higher DO (1.8 ml l-1)
than stations 20020 and 20040 from the same transect, where the DO levels recorded were 0.67
and 0.53 ml l$^{-1}$, respectively,  while the offshore stations 23070 in transect OWB 23$^0$ S and 26090



in transect OL $26^0$ S recorded the highest dissolved oxygen (2.30 and 4.34 ml l$^{-1}$, respectively)
(Table 1). Fine sand was the most common sediment size ranging between 30-38.8% in all stations.
Most stations from the sites also recorded higher proportions of medium sand and silt, except for
station 23020 in transect OWB $23^0$ S, which recorded a lower proportion of silt and an increased
abundance of coarse sand compared to the other stations (Table 2).
**3.2 Biotic factors**
Stepwise regression analysis revealed that most diversity indices (H' Diversity, Evenness,
Dominance, Berger-Parker, Fisher Alpha, Equitability J, and Brillouin) had no predictive variables
except Margalef index, Richness, and Menhinick whose predictors were identified as TOM, DO,
and very coarse sand respectively. Pearson correlation analysis indicated a significantly high
correlation between DO and TOM, as TOM reflects surface production, which is the driver of the
low DO at the BUS (Table S1).
Due to the lack of replicates in some stations and the high predictive role in DO compared to the
other two factors, the sampling stations were thus grouped based on their recorded levels of DO
as described by (Levin Lisa, 2003), i.e., stations with DO less than 0.1 ml l$^{-1}$ were grouped as
'Microxic,' those with DO between 0.1 and 1.0 ml l$^{-1}$ grouped as 'Dysoxic' while those with DO of
1.0 ml l$^{-1}$ and above grouped as 'Oxic' in a bid to analyze how DO affects macrobenthic
communities structure.
**3.3 Macrobenthic assemblages**
Macrofaunal densities differed significantly across the various oxygen zones; the Microxic stations
recorded very low densities (4,661±4,834 ind. m$^{-2}$) and had the lowest number of taxa with only



six taxa present. The Dysoxic stations, on the other hand, recorded the highest densities
(74,108±134,126 ind. m$^{-2}$), with one station from this Oxygen range (20040) recording an
extremely high density of 274,991 ind. m$^{-2}$ boosted by a high abundance of Nematoda and
Oligochaeta but with a low number of taxa (S=6). The Oxic stations recorded the highest number
of taxa (18) and the second-highest average density but were comparatively lower than those
registered by the dysoxic stations (14,345±6,726 ind. m$^{-2}$) (Fig. 2).
A total of 19 macrobenthic taxa were identified and were dominated by Nematoda, Polychaeta,
Oligochaeta, and Cumacea, the only peracarid crustacean abundant in all the oxygen zones.
Microxic stations recorded the lowest taxa count (S=6), which constituted Cumacea (37.5%),
Polychaeta (26.9%), Nematoda, Oligochaeta, and Ascidia as the only taxa present in the Microxic
stations (Fig 3).
Although the taxa count in the dysoxic stations was like that of the microxic stations, the
composition, and dominance were different. The macro-nematoda was the most dominant taxa in
this zone, outnumbering all the other taxa. Its numbers were very high in one of the stations
(233,354 ind. m$^{-2}$). Oligochaeta was the second most abundant taxon in this oxygen zone, with its
high abundance coming from the same station with the highest nematode counts. Polychaeta and
Cumacea were the dysoxic stations' third and fourth most abundant taxa. Bivalvia and Ostracoda,
which were absent in the microxic stations, were present in the dysoxic stations, albeit in low
abundance (Figure 3).
The oxic stations recorded the highest number of taxa counts, with 18 out of the 19 taxa recorded
overall. Taxa like Echinodermata (Ophiuroidea), Holothuroidea, Aeolosomatidea, Isopoda,
Aplacophora, and Amphipoda were recorded in this zone with average abundances of more than



1%. Nemertina, Turbellaria, and Cnidaria were also recorded in these stations; however, their
abundances were below 1%, and they were grouped as others (Figure 3).
All the oxygen zones were dissimilar to one another. The highest dissimilarity was observed
between the Dysoxic and Microxic zones which were 77.99% dissimilar despite both stations
being characterized by low dissolved oxygen levels. The Oxic stations were also highly dissimilar
to the Microxic and Dysoxic zones, with values of 68.58% and 65.91%, respectively (Figure 4).
The Dysoxic and Microxic stations recorded low macrobenthic H' diversity (<1), with the Microxic
stations recording slightly higher diversity indices than the Dysoxic stations, while the Oxic
stations recorded the highest H' diversity (1.46±0.4). The dominance followed a negative trend to
the H' diversity as the Dysoxic stations recorded the highest dominance (0.6±0.28). In contrast, the
Oxic stations recorded the lowest dominance (0.31±0.18), while the Microxic Dominance index
fell in between the two (0.5±0.18) (Figure 5).
**3.4 Macro-nematodes density and diversity**
On average, macro-nematodes were the most dominant taxon in this study as a result of their
dominance in the dysoxic station. Macro-nematodes were abundant in all oxygen zones recording
relative abundances of 8%, 74%, and 24% in Microxic, Dysoxic, and Oxic zones respectively.
This meant that they had a substantive contribution to the macrobenthic densities as they were
621±879 nematodes $m^{-2}$ from a total of 4,661±4,834 ind. $m^{-2}$ in the Microxic zone,
61,912±114,424 nematodes $m^{-2}$ in the dysoxic zone out of 74,108±134,126 ind. $m^{-2}$, and from
14,345±6,726 ind. $m^{-2}$ in the Oxic zone 4,454±2,906 were nematodes.





Eighteen genera of macro-nematodes were identified. *Desmolaimus* and *Paracomesoma* were
present across all stations, with their abundances peaking in Dysoxic stations (together with
*Metoncholaimus*), and were the only genera recorded from the Microxic zone.
*Paralongicyatholaimus* and *Neochromadora* recorded significant relative abundance (>4%) in
Dysoxic stations. In contrast, *Thalassolaimus, Paramesacanthion, Enoploides, Halanonchus,*
*Rhabdodemania,* and *Dorylaimopsis* recorded significant abundance in Oxic stations but were
absent in Dysoxic stations except for *Thalassolaimus*. *Metoncholaimus* and
*Paralongicyatholaimus,* were present in Dysoxic stations but absent in Oxic stations, while
*Paramesacanthion, Enoploides, and Rhabdodemania* were present in the Oxic station and absent
in the Dysoxic stations. Graphing the relative abundance, *Thoracostomopsis, Anticoma,*
*Cephalanticoma, Trileptium, Mesacanthoides, Terschellingia,* and *Marylinnia* were grouped as
others as they recorded insignificant abundances (<4%) and were absent in Dysoxic station except
for *Marylinnia* and *Terschellingia,* whereby, the former was absent in the Oxic station while the
latter was present in both oxygen zones  (Fig. 6).
The feeding guild differed between the various oxygen zones identified. Epistratum feeders
dominated the dysoxic zones (62%), followed by predators/omnivores (28%), and finally, selective
deposit feeders (10%). On the oxic zone, selective deposit feeders were the most dominant feeding
guild (56%), while epistratum feeders, predators, and omnivores had the same abundance of 22%
each. No non-selective deposit feeders were recorded in this study (Figure 7).
The macro-Nematoda diversity portrayed a similar trend as the general macrofaunal diversity, with
one of the Microxic stations (23002) lacking nematodes and its partner station (23020) registering
only two nematodes therefore, it was left out during diversity analysis. The remaining oxygen
ranges (Dysoxic and Oxic) portrayed the same trend as the general macrofaunal trend within the



study site as the Oxic stations recorded higher H' Diversity (1.38±0.5) than the Dysoxic
(0.81±0.84) stations. In contrast, the Dysoxic stations recorded higher dominance (0.59±0.39) than
the Oxic stations (0.32±0.18) (Fig. 8).
**4.0 Discussion**
Upwelling systems are known for their high surface productivity and Oxygen Minimum Zones
(OMZ), which impinge on the benthic zone creating strong oxygen gradients on the seafloor and
acting as the dominant driver for benthos diversity in these zones (Zettler et al., 2013). Despite the
Benguela Upwelling System (BUS) being recognized as one of the major Eastern Boundary
Upwelling systems, there is limited information on the structure and composition of the benthic
communities. Information on macrofauna communities in the BUS will not only improve the
existing database on benthic fauna but also provide insight into how increasing hypoxic areas in
the ocean might structure benthic communities.
In this study, oxygen correlated significantly with various diversity indices; thus, the different
stations were grouped into three based on the oxygen levels recorded. The structure and
composition of the macrofauna communities varied among the various oxygen zones. Most
macrofaunal studies identify Polychaeta as the most abundant macrofauna taxon in both oxic and
hypoxic areas (Eisenbarth & Zettler, 2016; Soto et al., 2017). However, in this study, Polychaeta
only had the highest relative abundance in the Oxic zones. In the microxic zone, the relative
abundance was dominated by Cumacea, followed by Polychaeta. It is essential to note that
numerically Polychaeta was the most abundant in this oxygen zone, but the presence of other taxa
in these stations reduced their relative abundance. The presence of Cumacea in high quantities in
the core OMZ has been reported by Zettler et al., (2013) and Eisenbarth & Zettler (2016), who



described them as possible opportunistic species colonizing permanent hypoxic areas from
adjacent areas, and thus their abundance may be season-specific. Currie et al. (2018) attributed the
presence of Cumacea and other macrofauna taxa in the BUS to the Sulfur-oxidizing bacteria,
possibly providing a detoxified condition in this area. In this case, the mobility of the Cumacean
gives them an advantage over other tolerant taxa like polychaetes and nematodes at the core and
hence their high relative abundances at the OMZs core.
Some polychaete families have physiological adaptations to tolerate the low oxygen quantities
found in OMZs (Hanz et al., 2019; Joydas & Damodaran, 2014; Levin et al., 2009). At the microxic
zone, polychaetes had the numerical abundance in the microxic stations despite Cumacean leading
in relative abundance. The station where Cumacea were located (23002) had only two taxa with
Cumacea recording (75%) but with only 932 cumaceans $m^{-2}$, while the polychaetes recorded 4350
polychaetes $m^{-2}$ in the other microxic station (23020) but recorded a relative abundance of 53%
and hence its average lower relative abundance compared to the Cumacea.
In the dysoxic zone, Nematoda was the most dominant taxa outcompeting the Polychaeta in all the
dysoxic stations, recording more than 70% relative abundance. Oxygen can cause shifts in
community structure and trophic transfer (Neira et al., 2018), as evidenced in this study. Nematoda
as a taxon has not received significant attention in the macrofaunal size range in most studies
despite evidence of their presence therein (Joydas & Damodaran, 2014; Sharma et al., 2011). An
increase in the size of nematodes to macrobenthic class sizes has been reported in chemosynthetic
environments that experience similar characteristics as OMZs, i.e., low oxygen and high sulfidic
contents (Vanreusel et al., 2010). Apart from the increase in size, OMZs also tend to enhance the
regional dominance of tolerant organisms such as nematodes with high biomass recorded in
response to organic matter inputs. The high abundances are thought to reflect the availability of





organic matter, a significant nutrient source for macrofauna, coupled with release from predation
from larger fauna affected by the reduced oxygen concentrations (Moens et al., 2013).
The high nematode abundance in partnership with Oligochaeta occurred in the dysoxic zone;
oxygen conditions were low enough to exclude some taxa but sufficient for tolerant species to
survive and reproduce. Such conditions are referred to as the 'edge effect,' and such high densities
are characteristics of the edge of the OMZs, where various species have been observed to have
abnormally high densities (Moens et al., 2013; Neira et al., 2018). The reasons for these high
abundances are not well understood, but (Gutiérrez et al., 2008) alluded that the nematodes'
population can multiply in low oxygen conditions, which experience high loads of organic matter
input. Despite their tolerance to anoxia, nematodes cannot survive long-term exposure, as observed
in the anoxic zone. Thus, the patchiness and high variability in the dysoxic zone calls for more
studies (Buhl-Mortensen et al., 2010).
Once the DO levels increased to dysoxic levels, other taxa like ostracoda and Bivalvia were
observed, albeit in meager numbers. Despite ostracodes flourishing better in well-oxygenated
marine areas, various families (*Platycopina*) have been observed to tolerate and thrive in Oxygen
Minimum Zones. When the DO levels increased to above 1.0 ml $l^{-1}$, more taxa were recorded, and
these numbers increased more when DO was above 2.0 ml $l^{-1}$.
A general trend has been observed in the various studies in OMZs whereby macrofaunal species
richness and diversity reduce towards the core and increase as one moves away from the core
(McClain & Schlacher, 2015). Similarly, our core area (microxic) had the lowest density, diversity,
and species richness, with only 1243 ind. $m^{-2}$ recorded per core. A similar number was recorded
by Zettler et al., (2009). Once the Oxygen levels increased above 1 ml $l^{-1}$, more taxa were recorded,



and the high dominance of the tolerant taxa, as evidenced in the microxic and dysoxic areas, was
reduced. Taxa such as Amphipoda, Isopoda, Echinodermata, Nemertina, Aeolosomatidae,
Aplacophora, Holothuroidea, and Cnidaria were only recorded in the oxic zones indicating low
tolerance to low oxygen levels. Most of these fauna are crustaceans which Soto et al. (2017) also
recorded abundances in oxic stations. However, Zettler et al. (2009) recorded amphipod species in
the low oxygen areas indicative of species-specific tolerance/intolerance to hypoxia.
At the meiofaunal level (0.0038-1 mm), nematodes dominate with very high abundances in OMZs,
and their structure and composition have been well documented (Gutiérrez et al., 2008; Neira et
al., 2018). In contrast, despite various studies in OMZs acknowledging the presence of large
nematodes (>1.00 mm), little attention has been accorded to analyzing them further. In this study,
we analyzed macro-Nematoda to the genus level because of their dominance in the dysoxic zone
to understand and acknowledge the structure and composition of macro-Nematoda in the BUS and
OMZs in general.
Macro-Nematoda abundance varied across the OMZ, with very low abundance in the microxic
zones, extremely high numbers at the dysoxic zones, and a substantial amount at the oxic stations,
accounting for only 28% of the total abundance. Nematodes are considered one of the most tolerant
taxa in the marine environment, with the ability to tolerate low oxygen and high sulphidic
environments characteristic of OMZs and may reach very high abundances in these environments
(Ingels et al., 2023). This was the case at the dysoxic zone, where nematodes recorded high
abnormal densities in one station, indicating a high tolerance of these taxa and the ability to grow
to large sizes and even dominate the macrofaunal component. Even with such high tolerance levels,
Nematoda abundance can be impacted by microxic conditions, as observed in the microxic areas
with a recording of only 621 nematodes m$^{-2}$ in this oxygen zone. This meager value, however, may



be at the macrofauna level, and the case may be different at the meiofauna level, where nematodes
have recorded substantial densities in microxic environments (Neira et al., 2018; Steyaert et al.,

336    2007).

Despite their high abundance in OMZs, not all nematodes are tolerant to low oxygen levels (Moens
et al., 2013), as observed in this study. *Metoncholaimus, Paracomesoma,* and *Desmolaimus*
dominated the dysoxic zone; these three genera are members of *Oncholaimidae, Comesomatidae,*
and *Linhomoeidae*, respectively. Members of the family *Oncholaimidae* have large bodies that can
disperse rapidly and colonize carcasses of macrofauna and even fish that may have succumbed to
the low levels of oxygen found in the dysoxic zone. Nevertheless, their bodies are large enough to
fit within the macrofauna size range, while their ability to swim ensures they actively locate their
food source (Moens et al., 2013). Their high abundance in this study might reflect a congregation
upon a food source that had attracted nearby members in large numbers. Their ability to colonize
the 'food source' in such numbers in a dysoxic environment indicates their tolerance to low oxygen
levels.
On the other hand, *Comesomatidae* and *Linhomoeidae* members have been noted to have high
abundances in enriched sediments with low oxygen, indicating tolerance to anoxic conditions
(Steyaert et al., 2007). Their long and slender bodies might be the reason for their records at the
macrofauna level. Despite this generalization at the family level and the assumptions that members
of the same family may portray similar life strategies (Bongers et al., 1991), tolerance of nematodes
to hypoxia is species-specific (Moens et al., 2013) as Steyaert et al. (2007) observed members of
the same genera (*Sabatiera*) reacting differently to hypoxic and anoxic conditions. Thus, further
analysis should be done to identify the species that are tolerant to hypoxia at these OMZs at
macrofauna levels.



Tolerance is determined by both the absence and presence of taxa; most genera present in the oxic
zone were absent in the dysoxic area and may be seen as genera intolerant to low oxygen levels.
Ridall and Ingels (2021) categorized *Anticoma* as an indicator of hypoxia due to its intolerance to
hypoxia.
Weiser's feeding types have long been used to assess the trophic structure of nematode
communities. This study had a high abundance of selective feeders (1A) and epistratum feeders
(1B), with the latter dominating the oxic zone. Neira et al., (2013) recorded a complete dominance
of selective deposit feeders in one of the shallow OMZs stations and a dominance of both the
selective deposit feeders and epistratum feeders on the other station with a slight abundance of the
non-selective deposit feeders (1B).
This appears to be among the exceptions to the general rule that non-selective deposit feeders
dominate substrates with a high abundance of organic matter, as the opposite trend was observed.
From this study and Neira et al., (2013), the classification by Moens & Vincx, (1997) would
provide insight into explaining the trophic structure. They modified the Wieser's deposit feeders
from selectivity (due to lack of knowledge on selectivity) to their diet types as either microvores,
ciliate feeders, or deposit feeders. As the reduced oxygen in OMZs negatively impacts the
abundances of multicellular organisms, the role of microbes in such areas becomes prominent
(Dietrich et al., 2021), which may favor the abundance of microvores tolerant to hypoxia over
deposit feeders. The areas around the OMZ have also been observed to contain a high level of
diatomaceous mud, which forms a significant component of epistratum feeders' diet, whose
abundance was also high in the dysoxic areas. Below the OMZ, the production of diatoms is
reduced, and the abundance of epistratum feeders also reduces, giving rise to the dominance of
selective deposit feeders.





**5.0 Conclusion**


In summary, the Benguela Upwelling System (BUS) is a major Western Boundary Upwelling
system known for its high surface productivity and Oxygen Minimum Zones (OMZ). This study
found that the structure and composition of macrofauna communities in the BUS varied among
different oxygen zones, with Polychaeta being the most abundant macrofauna taxon in oxic zones,
Cumacea dominating in microxic zones, Nematoda being the most dominant in dysoxic zones, and
Ostracoda and Bivalvia observed limited numbers in the anoxic zone. These results suggest that
oxygen levels significantly shape benthic communities, with tolerant species dominating in low-
oxygen environments and thus the increasing hypoxic areas in our ocean might result in reduced
macrobenthic faunal densities, diversity, and species richness. Further studies are needed to
understand the mechanisms behind the observed patchiness and high variability in the dysoxic
zones.












### 6.0 Tables

*Table 1: Station information and abiotic factors information recorded from the stations in the Northern Benguela Upwelling System*

| Transect | Station | Longitude | Latitude | Depth (m) | TOM(%) | DO (ml l$^{-1}$) |
|---|---|---|---|---|---|---|
| Off Cape Frio (OCF-20$^0$S) | 20002 | 12.99905 | -20 | 33 | 27.47±1.55 | 1.8 |
| | 20020 | 12.67858 | -20 | 125 | 38.64±2.16 | 0.67 |
| | 20040 | 11.79321 | -20 | 219 | 17.58 nr | 0.53 |
| Off Walvis Bay (OWB-23$^0$S) | 23002 | 14.3734 | -23 | 39 | 23.08 nr | 0.06 |
| | 23020 | 14.06986 | -23 | 128 | 20.88 nr | 0.02 |
| | 23070 | 13.14 | -23 | 318 | 9.89±1.55 | 2.3 |
| Off Luderitz (OL 26$^0$S) | 26090 | 13.28 | -26 | 1282 | 4.4 nr | 4.34 |





404

405    *Table 2: Sediment size distribution in the Northern Benguela Upwelling System*

| Transect | Station | Very Coarse Sand | Coarse Sand | Medium sand | fine sand | very fine sand | Silt |
|---|---|---|---|---|---|---|---|
| Off Cape Frio ($20^0$ S) | 20002 | 0.65 | 0.67 | 13.81 | 34.82 | 21.29 | 28.76 |
| | 20020 | 0.93 | 0.62 | 28.17 | 30.89 | 12.98 | 26.42 |
| | 20040 | 2.71 | 4.37 | 18.55 | 33.33 | 17.04 | 23.98 |
| Off Walvis Bay ($23^0$ S) | 23002 | 0.00 | 0.28 | 18.74 | 36.17 | 17.25 | 27.57 |
| | 23020 | 0.98 | 6.85 | 14.91 | 38.88 | 18.83 | 19.56 |
| | 23070 | 0.00 | 0.00 | 13.92 | 33.73 | 20.29 | 32.06 |
| Off Luderitz ($26^0$ S) | 26090 | 0.00 | 0.29 | 23.99 | 30.46 | 19.25 | 26.01 |

406





**7.0 Figures**

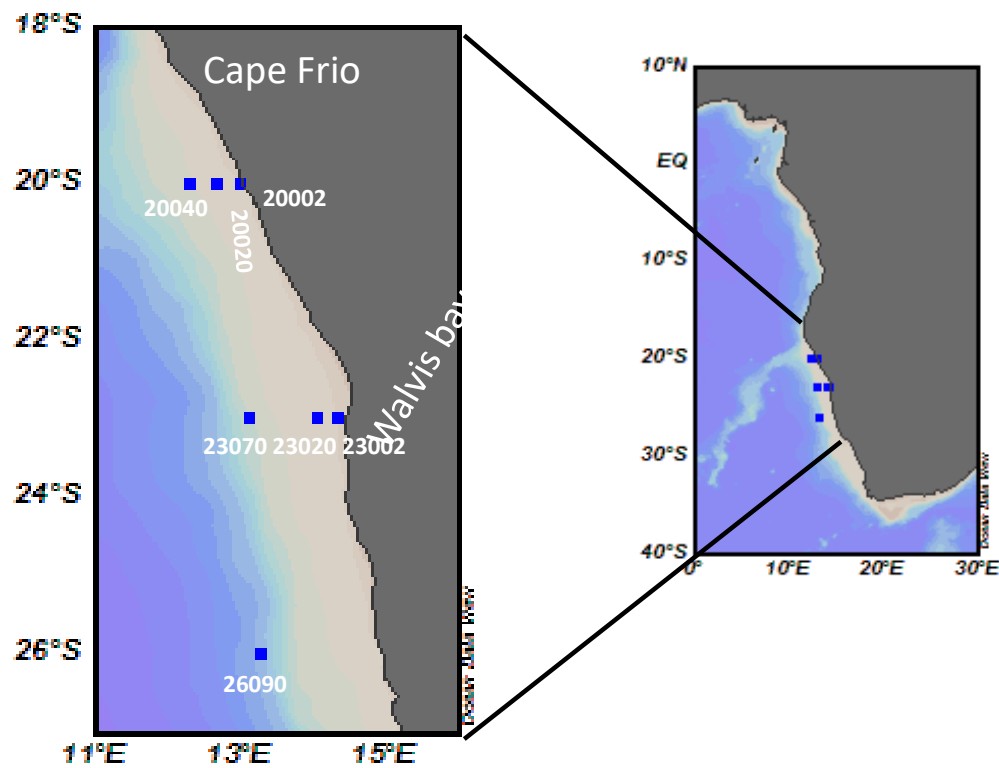

*Figure 1: Sampling stations located across the Northern Benguela Upwelling System*





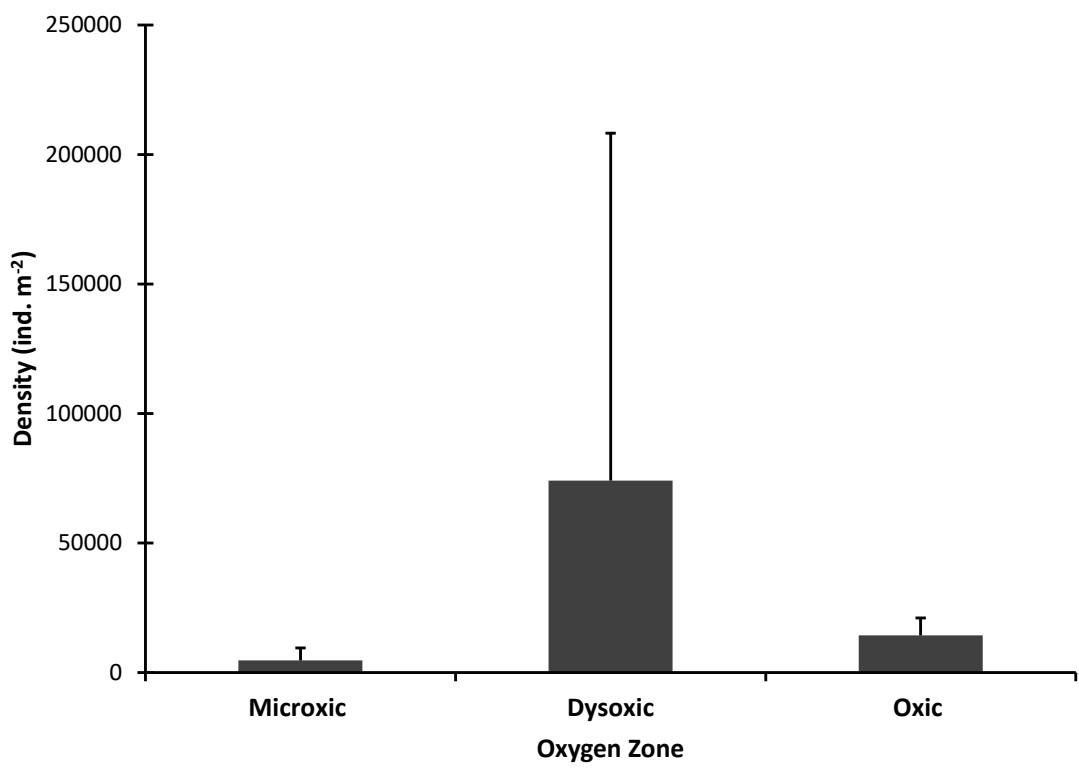

Figure 2: Macrobenthic densities in the different oxygen zones in the Northern Benguela Upwelling system





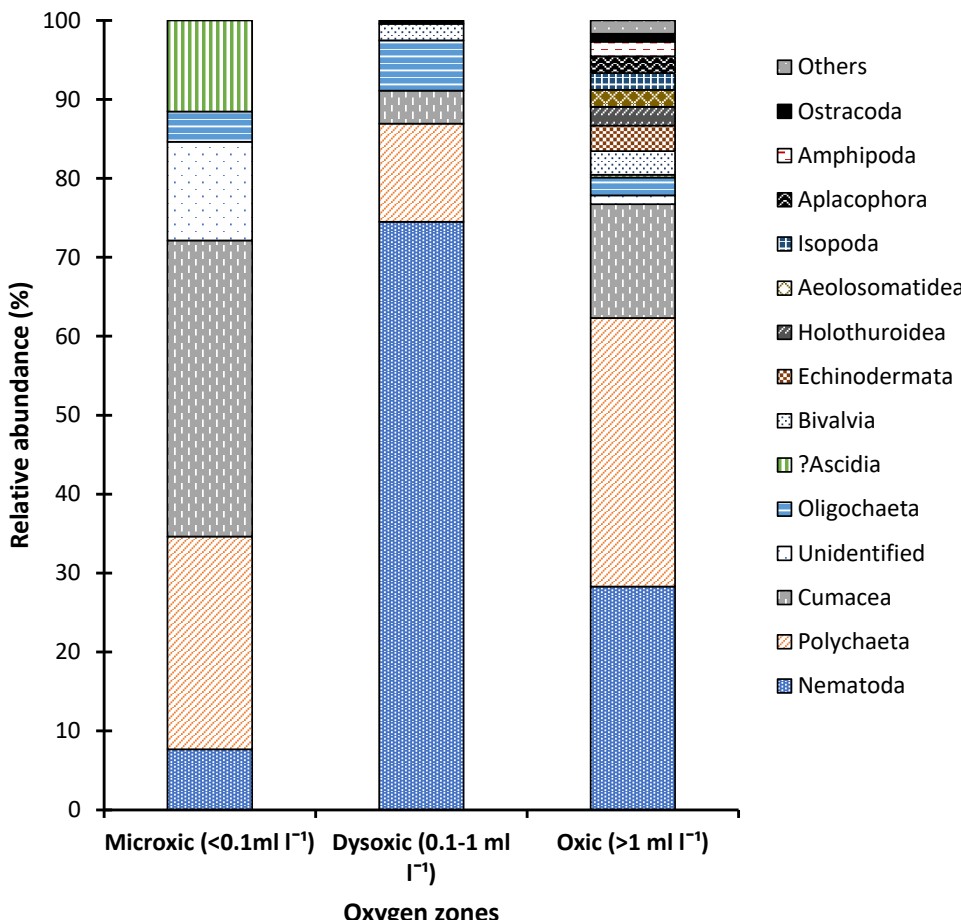

Figure 3: Macrofauna Relative abundance in the Northern Benguela Upwelling System, based on the oxygen zones identified.





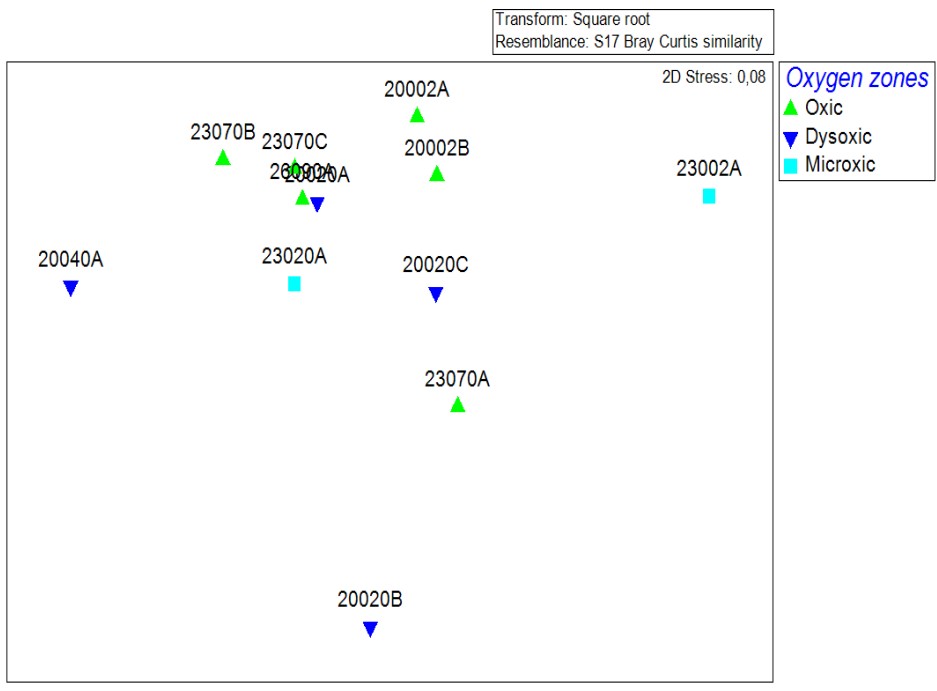

Figure 4: Non-metric multidimensional scaling (nMDS) plot based on Bray-Curtis Similarity index of macrobenthic fauna communities recorded in the Northern Benguela Upwelling System.





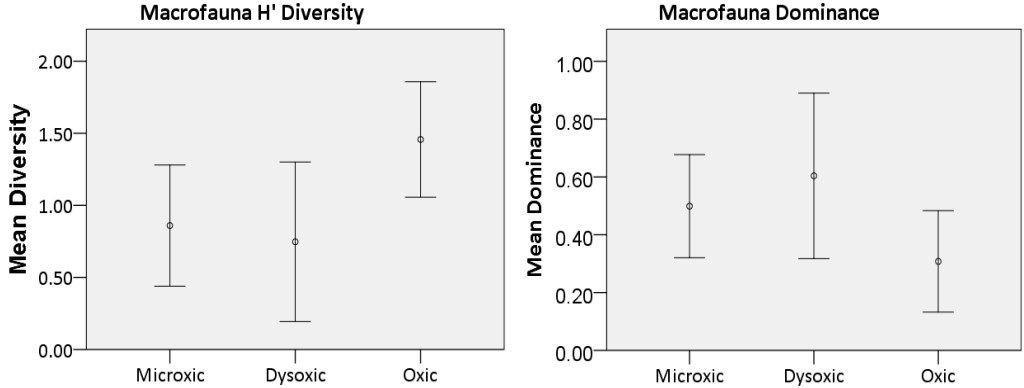

Figure 5: Macrofauna diversity indices recorded in the various oxygen zones in the Northern Benguela Upwelling system.





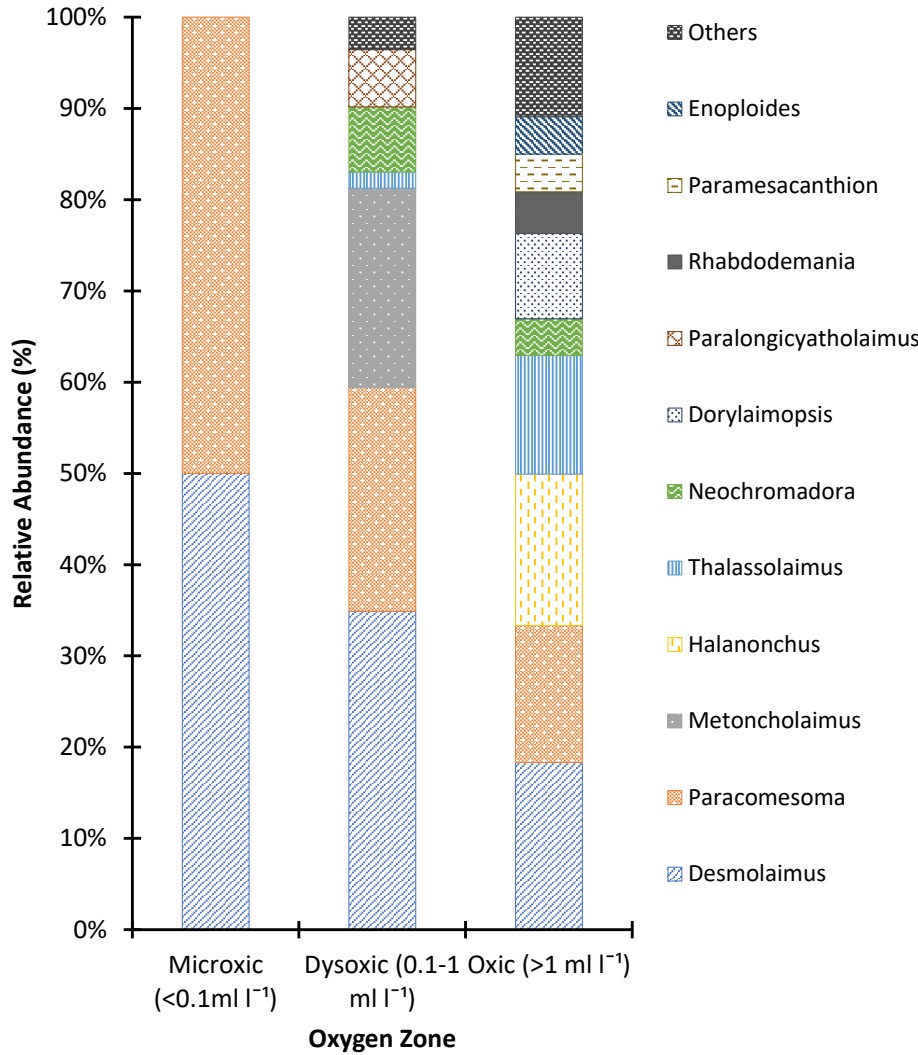

Figure 6: Macronematoda Relative abundance in the Northern Benguela Upwelling System, based on the oxygen zones identified.



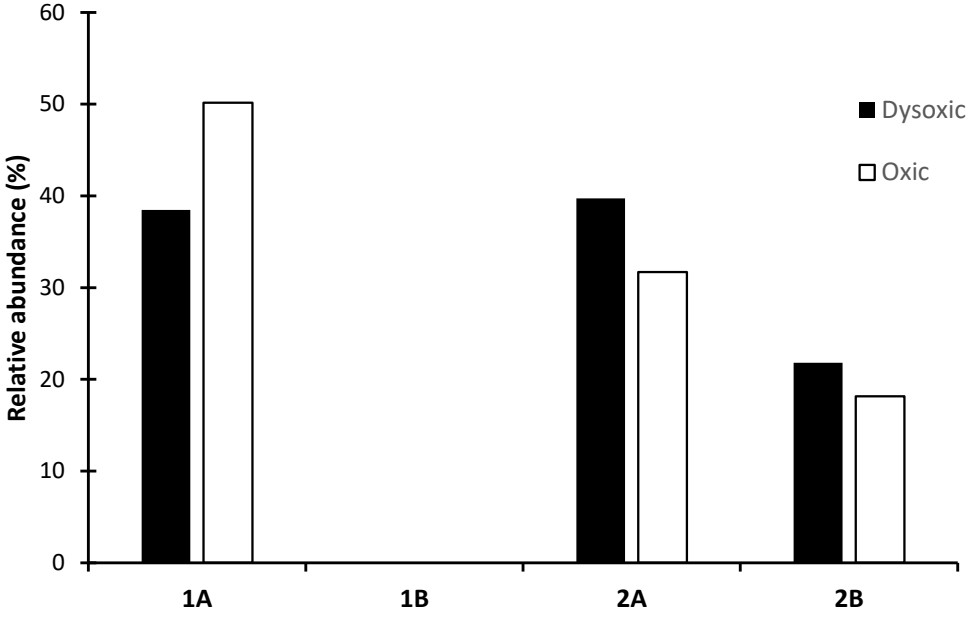

Figure 7: Feeding guilds based on Wieser of macro-Nematoda from the Northern Benguela Upwelling System. (1A= Selective deposit feeder, 1B- Non-Selective deposit feeder, 2A=Epigrwoth feeders, 2B=Predators/Omnivores).





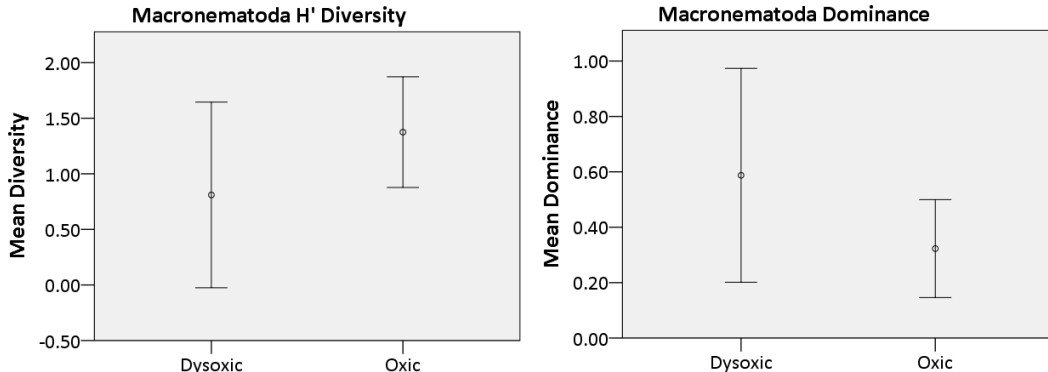

Figure 8: Macro-Nematoda diversity indices recorded in the various oxygen zones in the Northern Benguela Upwelling system.

**8.0 Code/Data Availability**

Currently, the data is not available but can be released upon request to the authors.

## 9.0 Author contribution

HS was responsible for the conceptualization of the study. HS and WB conducted the investigation, formal analysis, and drafting of the original draft. MA was responsible for supervision, review, and editing of the final draft.

## 10.0 Competing interests

The authors declare that they have no known competing financial interests or personal relationships that could have appeared to influence the work reported in this paper.

## 11.0 Special Issue Statement

Part of this manuscript was presented as a poster presentation during the 53$^{rd}$ International Colloquium on Ocean Dynamics: 3rd GO2NE Oxygen Conference held in Liege, 2022. As a result, the corresponding author was invited to submit a manuscript for the special issue *"Low-oxygen environments and deoxygenation in open and coastal marine waters"* as part of the journal Biogeosciences.

## 12.0 Acknowledgements

The authors would like to thank the 3rd Regional Research Graduate Network in Oceanography (RGNO) organizers, all the sponsors (, and participants for technically and financially supporting the authors in the participation and subsequent sampling that resulted in the successful completion of this study. The assistance from the Namibian National Marine Information and Research Centre



(NatMIRC) was instrumental in successfully sampling, processing, and transporting samples. The crew of the sampling vessel R/V Mirabilis ensured a smooth sampling experience, and hence we take this opportunity to appreciate their services.





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
