# Peer review of "Influence of Oxygen Minimum Zone on Macrobenthic Community Structure in the Northern Benguela Upwelling System: A Macro-Nematode Perspective"

_Biogeosciences, 2023_

## Referee Comment (RC1)

**Influence of Oxygen Minimum Zone on Macrobenthic Community Structure in the Northern Benguela Upwelling System: A Macro-Nematode Perspective**

Hashim Said Mohamed, Beth Wangui Waweru, Agnes Muthumbi

Peer Review – 15/11/2023

**Review Summary**

This paper provides new data regarding the benthic communities of oxygen minimum zones (in this case, the Northern Benguella Upwelling System), a generally under-studied habitat, and, as such is within the scope of BG. Unlike most macrofauna studies, the macro-nematodes (>0.5 mm) are included in the macrofauna analyses which provides a novel perspective.

The taxonomic discrimination of the general macrofauna was undertaken to 'major taxa' level whilst the macro-nematodes were identified to genus level. This allows for general comparisons with other studies to be undertaken and key patterns in macrofauna and macro-nematode distribution and hypoxia tolerances to be indicated and discussed.

The fact that the samples were sieved on 0.5 mm mesh precludes the entire nematofauna being considered (the majority of which fall within the 'meiofauna' category). This is, however, acknowledged and the term macro-nematodes is used correctly to distinguish these large species from the nematode community as a whole.

The text is, at times, confusing to the reader unfamiliar with the study and study site, and needs revision to enable an easier read. I have listed comments regarding this below along with any formatting errors/queries I have noticed.

The only significant scientific query I have is regarding the nematode feeding type categories applied to the various genera and, therefore, the conclusions drawn from these. This needs clarification and justification with references.

I think this paper should be published following revision and will complement the 'low-oxygen environments and deoxygenation in open and coastal marine waters' BG special issue.

**Key Revisions and Clarifications Requested**

☐ The data analyses undertaken to justify the separation of the sites into the microxic, dysoxic and oxic zones (i.e. the SPSS) presents a trip-step in the results section. Consideration should be given to stating in the methods that these zone definitions were adopted (following Levin 2003) and removing section 3.2 from the results section. See comment for lines 130-134 and 162-174 below.

☐ Clarification required regarding the Weiser feeding type allocation with justification of the allocation of *Desmolaimus* and *Halanonchus* to Type 2A. If these genera are subsequently considered Type 1B, the results and discussion will need revising accordingly.

☐ Consideration should be given to including a fauna results summary table to include sample/site and community statistics (number of taxa, densities etc) that can be referred to when reading the narrative.

☐ Further comments, queries and suggestions are listed below.

***Text and Presentation Comments***

Line 34 – Reference required for hypoxia/SDG statement.

More detail on the study location required for reader unfamiliar with the Namibian coast and the BUS:

> Line 46 – Location details required. E.g. *The Benguela upwelling system (BUS) is located off the West African coast extending from……….*

> Line 62 - Location details required. E.g. …….*(namely, Walvis Bay, Namibia (the location of this study), California,…….)*

> Line 74 – Kunene river location details required. Reference to Figure 1 map with Kunene River labelled?

Lines 85/86 – reference to Figure 1 map required.

Figure 1 – Needs more annotation for readers unfamiliar with the area. Small map needs country boundaries and names. Large map requires additional locations, namely, Luderitz and Kunene River, as these are mentioned in text.

Figure 1 – needs a colour scale bar or key for depth as this is mentioned in the site description (lines 85-89)

Line 94 – clarify whether '*high surface primary production*' is mud surface or ocean surface. Obvious maybe, but avoid ambiguity.

Line 101 – reference to Figure 1 required after *at 90 nm*.

Line 103 - reference to Figure 1 not required. Reference Table 1 instead??

Line 114-118 – please provide details of the level of taxonomy for non-nematode macrofauna taxa. Later, the term 'macrofauna taxa' is used without the reader knowing the definition of 'taxa'.

Line 119 – Need to state that feeding types were ascribed to genera following Weiser (1953) and insert reference in ref list.

Line 130-144 – As above, it is not clear what level of taxonomic discrimination was used for statistical analyses.

Line 130-134 – This needs clarifying. Something along the lines of…..

*Out of the seven sampled stations, only three had replicates. Therefore, direct statistical comparisons between sampling stations was not possible. To overcome this, the sampling stations were grouped into 'habitat types', based on the measured environmental variable that provided the most significant correlation with the various biotic indices. The communities within these key habitat types were then compared using indices and analyses outlined below.*

An alternative approach that may be less complicated would be to state up front in this section that, following the approach of Levin (2003), the samples from microxic, dysoxic and oxic habitats will be compared in order to determine the effects of low DO. It could be further stated that this approach was justified by the application of SPSS that indicated DO was a key determinant associated with the microbenthic community structure. The text from lines 169-174 could be modified for this purpose.

Line 134 – New paragraph starting at *The data recorded*….

Line 142 – *the various replicates were considered as a single samples*. How was this done? If the replicate data were added together it would constitute 3 times the search effort as that for a single sample, therefore, the number of taxa will likely be elevated. Perhaps a more valid approach would be to randomly select a single replica for use in this analysis, rather than combine the three replicates?

Line 147 – suggest term *inverse relationship* in preference to *opposite trend*.

Line 147-150 – not easy to follow. Suggest

*Total Organic Matter (%TOM) demonstrated an inverse relationship with depth, with higher organic matter values recorded in the shallower stations. For example, the shallowest stations 20020, 20002, and 23002 had the highest TOM of 38.6%±2.16, 27.5%±1.55 and 23.1%, respectively, whilst the lowest TOM values were found in the most offshore stations, stations 23070 and 26090, with 9.89%±1.55 and 4.4%, respectively (Table1).*

Line 151 - *Similarly, dissolved oxygen values had lower values recorded in the shallower onshore station* – According to Table 1, this is not correct; needs clarifying or removing. The OFC transect had its highest DO at the shallowest site (33 m) and OWB had lowest DO at middle depth site (128 m), as described in the later text.

Line 158 – New paragraph after *(Table 1)*.

Lines 162-174 (3.2 Biotic Factors) – As discussed for Lines 130-134 above, it is suggested that the use of the approach of considering the sampling stations in terms of their DO zones (following Levin, 2003) be stated up front in the methods section. This section is somewhat of a trip-step in the narrative of the results.

Line 171 – reference error. Should read *(Levin, 2003)*.

Line 162 – Section 3.2. As discussed above for Lines 130-134, it may be easier to follow if this narrative is moved to the data analyses methods section 2.4, to justify using the three DO habitat types.

Line 175 – 3.3 Macrobenthic assemblages. The term taxa is used with no definition (as mentioned for lines 114-118, above). Presumably, this is the taxonomic groups as listed on Figure 3, although the rarer are combined as others. Maybe list the taxa in a table?

Line 176-183 – This paragraph discusses the densities. The references to number of taxa are repeated in the subsequent paragraphs. Suggest the references to number of taxa are removed from this paragraph.

Line 215 – Insert reference to Figure 3, i.e. …… *zones, respectively (see Figure 3)*.

Line 216-219 – list of data is probably unnecessary as % abundance data already quoted. If a results summary table was included, this could be referred to instead.

Line 222 – …..*their abundance peaking in the Dysoxic station*….. Is this relative abundances or total densities. If relative abundance, please state so and reference Figure 6 at the end of sentence.

Line 226 - …..*recorded significant abundance*……. If this is not a reference to statistical significance, please use alternative term, e.g. high abundance, to avoid ambiguity. Similarly, in line 232, please use *low abundance* unless statistically insignificant.

Line 230-234 – sentence long and complicated. Need clarifying and/or shortening. Suggest:

*For the purposes of graphing the relative abundances, the genera Thoracostomopsis, Anticoma, Cephalanticoma, Trileptium, Mesacanthoides, Terschellingia, and Marylinnia were grouped as 'others' as they were recorded at low abundances (<4%); see Fig. 6.*

Line 235-237 – For clarity, please include the feeding type code after the first mention of the feeding type, e.g. *epistratum feeders (type 2A).*

Line 239 – Clarification and justification is required regarding the allocation of feeding types. Whilst I appreciate that allocation of Weiser's feeding types is based on the morphology of the buccal cavities and is, therefore, partially subjective, most linhomoeid nematodes (with the exception of *Terchellingia*) are considered Type 1B (non-selective detritivore). I would classify *Desmolaimus* as Type 1B due to its unarmed (the cuticular 'arches' are unlikely to act as teeth) and relatively large buccal cavity. Similarly, I have always considered *Halanonchus* a Type 1B non-selective detritivore as, apart from buccal cuticular thickening, there is no dentition and has a large open mouth. Furthermore, these Genera tend to occur in muds. The results and discussion of feeding type guilds may need reviewing.

Line 256 – ….*oxygen correlated significantly with*….. Please state significance of correlation ($p$ value?)

Line 257 – Typographical error. Should read ………*into three zones based on*…..

Line 257 – for the benefit of the reader, it would be useful to remind of the three zones. Suggest inserting in parentheses, e.g. …….*oxygen levels recorded (microxic zone (<0.1ml l$^{-1}$); dysoxic zone (0.1-1.0 ml l$^{-1}$); oxic zone (>1.0 ml l$^{-1}$)).*

Line 262-264 – *It is essential to note*……….*their relative abundance*. Does this mean polychaetes were recorded at their highest density at the microxic sites or they were the most abundant taxa in this zone? I assume the former. Needs clarification to avoid ambiguity.

Line 265 – Please define the *core OMZ*. OMZ spatial centre? Area of lowest DO? Maybe in parentheses.

Line 273-275 – *At the microxic zone*……..*in relative abundance*. Meaning unclear. Does it mean, at the microxic sites, the polychaetes were the most abundant taxa in terms of total density, but the average relative abundance was lower than that of cumaceans? (How was the relative abundance of multiple sites calculated?)

Line 276 - Were there only 4 specimens found at 23002, i.e. 3 cumaceans (75%) and 1 other taxa (25%)? If so, the 75% relative abundance, whilst correct, gives a misleading impression of the community and, when averaged with the other microxic station could be misinterpreted.

- This paragraph needs reviewing as it is not clear what point is being made and it read more like the results narrative than discussion.

Lines 269, 274, 275 – The term cumacean should not be capitalised whereas Cumacea should. Please use terms consistently, either cumaceans or Cumacea wherever possible. Similarly, polychaetes or Polychaeta; nematodes or Nematoda etc. My preference would be polychaetes/nematodes etc when discussing the results, unless referring specifically to the phylum or class as a whole, in which case 'the Nematoda' or 'the Polychaeta' is good.

Line 279 – please use the term macro-Nematoda or macro-nematodes, throughout. The Nematoda includes the meiofaunal size category whereas, this study, considers only those retained on a 500 μm mesh, i.e. the macrofauna size class.

Line 286 – Should this read *Apart from the increase in nematode size*….

Line 289 – suggest ……*coupled with a reduction in predation by larger fauna that are affected adversely by reduced oxygen concentrations*…..

Line 296 – reference should read Gutierrez et al. (2008)

Line 301 – *Once the DO levels to dusoxic levels*…… This implies a temporal data set that documented an increase in DO. Suggest *At dysoxic sites (DO 0.1 – 1 ml l$^{1-}$), other taxa*……

Line 304 – as above. Suggest *At the oxic sites, where DO levels were above 1.0 ml l$^{-1}$, more taxa*……

Line 308 – Lowest densities, diversity and species of macrofauna taxa or macro-nematodes or both? Please clarify.

Line 310 – As 301 and 304 above. Suggest *At sites with DO above 1 ml l$^{-1}$*……..

- Does this refer to your study or Zettler's? Avoid ambiguity, suggest *During the present study, at sites with*….

Line 314-316 – *Most of* ………. *tolerance/intolerance to hypoxia*. Meaning unclear. Suggest: *Of these fauna, crustaceans were most abundant. This conforms to the observations of Soto et al. (2017) at oxic sites in an upwelling system in Chile. Conversely, Zettler et al. (2009) recorded amphipod species in low oxygen areas. These contradictory results indicate that, at least amongst the Amphipoda, tolerance/intolerance to hypoxia is species specific.*

Line 357 – Suggest: *Tolerance to hypoxia is indicated by both the presence and absence of taxa*……..

Line 361 – insert Wieser reference

Line 362-363 – Need consistence of terminology for feeding types in text and Figure 7. Standard definitions are:

| Type | Buccal Cavity | Feeding Method |
|------|---------------|----------------|
| 1A | Small, unarmed | Selective deposit feeders and bacteriovores |
| 1B | Large, unarmed | Non-selective deposit feeders |
| 2A | Small, armed | Selective epigrowth feeders and herbivores |
| 2B | Large, armed | Carnivores and omnivores |
| | | |

Line 363 - Typographical errors – I think it should read ……*(2A), with the former dominating the oxic zone.*

Line 367 – suggest *These observations appear to be*……

Line 375-376 - *The areas around the OMZ have also been observed to contain a high level of diatomaceous mud, which forms a significant component of epistratum feeders' diet, whose abundance was also high in the dysoxic areas.*

Is it true that epistratum feeders (2A species) feed on diatomaceous mud? Need reference. 2A species feed on live diatoms whereas diatomaceous mud comprised dead, settled diatoms. Does the surface of the mud support epibenthic diatoms? The shallowest site was 33 m (oxic) whilst the rest of the oxic and dysoxic sites were in excess of 100 m where primary production would be very limited.

Line 377 – *Below the OMZ,….* Does this mean offshore from the OMZ, in deeper water?

Line 385 – should this read *Ostracoda and Bivalvia observed in limited numbers in the oxic zone…..?* (not anoxic?)

Line 380-391 (5.0 Conclusion) – the conclusion does not mention any thoughts on the macro-nematodes despite the title stating "a macro-nematode perspective".

---

## Referee Comment (RC2)

Specific comments:

**Abstract**

Need to start with a sentence giving broader scientific context for your study.

Macrobenthic – use a small 'm', not capital 'M' (many similar mistakes in ms with Macrofauna, Macro-nematoda, Dominance (!) etc…). I suggest do the same for oxic, microxic and dysoxic

Instead of Macro-Nematoda, just use 'nematodes' once you have established that you are looking at macrofauna.

Line 18: 'were present' not 'recorded abundances' (correct throughout ms)

Line 21: 'were absent', not 'no abundance'

There is no concluding sentence at end of Abstract

**Introduction.**

Reverse order of 1st and 2nd sentence.

Line 45: need references at end of sentence.

Line 56: need more details on OMZ communities and their function.

Line 57-61 – too basic, delete. Just define macrofauna by short sentence in brackets when first mentioning it.

Line 67 – need reference at end of this sentence

Nematodes are barely mentioned in the Introduction but they are main taxon of interest. Need info on nematodes in general and in OMZ in particular

Line 73: specify where Walvis Bay is

Line 77: need reference at end of this sentence.

Line 78 – are there any other studies that can be cited in context of Namibian shelf?

Line 82: you do not mention the focus on nematodes.

**Methods**

Line 95 – how low are the oxygen concentrations?

Table 1 – need to include number of replicates for macrofauna samples at each station

Line 114: why was 0.45 mm chosen for mesh size? Usually it is 500 or 300 microns. Not ideal for comparing with other studies.

Line 130 – this info needs to be in Table 1

Lines 130-134 – not easy to understand what you mean here. Just say what you did and why. How were correlations conducted? What do you mean by 'highest correlation'? what variable was chosen? Text in lines 169-174 needs to move from Results to Methods.

Line 138: need details of data treatment eg data transformation etc. Much too brief. What is anosim for? Simper?

Line 141: list the predictor variables in text or table. What is 'BUS'??? What selection criterion did you use for your stepwise regression? R2? Aikaike?

Line 142-144 – I don't understand what you mean. You matched biotic and abiotic data at scale of site for the correlation analyses?

**Results**

Line 149: these are huge TOM values

Line 167: give R2 values and P values in text.

Before giving results of correlation analyses, describe the microbenthic assemblages first (section 3.3. before 3.2)

Line 176: what statistical test was applied? P value? Of course they will be different because that's how you defined the groups to begin with.

Reorganise 3.3 to describe each group of stations – one paragraph per group.

Line 197: not 'taxa counts', just 'taxa'

Line 202: This sentence is much too vague. Dissimilar in what respect? Multivariate community structure? Statistically significant? P value? Pairwise comparisons? Show anosim table.

Lines 216-219: delete, sentence confusing.

Line 234: nee dto do multivariate community structure analyses, as per the macrofauna taxa data.

Line 235: why did you look at feeding guilds? There is nothing about this in the Methods or Introduction.

Discusssion

Line 257: clarify where the groupings come from. You use a scheme previously published?

Line 264: 'abundance', not 'quantities'

Line 273-275: I don't understand. If a group has highest relative abundance then it has highest absolute abundance too.

Line 275-278: confusing sentence. Delete.

Line 286: Vanreusel et al. make no such statement as far as I know. Indicate where in paper they say this?

Line 299: reference needed at end of this sentence Following sentence too vague. Why does patchiness 'call for more studies'?

Line 302: 'low', not 'meager'

Line 303: families are not italicised.

Line 309: "1234 ind. m-2 recorded per core" makes no sense.

Line 317: delete brackets and text within. This whole paragraph just repeats same things mentioned before.

Line 329: you cite a review paper, you need to cite papers providing actual data.

Line 331: nematodes may be larger in some areas because the species are different. Unlikely to be because conditions give them ability to grow bigger.

Line 336: why would meiofaunal nematodes ddiffer from macrofaunal nematodes?

Line 339: families are not italicised

Line 342: need reference for this sentence

Line 343: nematodes do not swim!

Line 360: so what? Did you record Anticoma in your samples?

Line 361: Wieser

Line 367: re-write this paragraph. Outline your main findings, how they compare with previous findings, and interpret their meaning.

Line 381-2: delete this sentence.

Line 387: did you identify species or just genera? Are you talking about nematodes?

Line 389: ok, so overall your findings confirm what we know already or is there anything different?

---

## Author Comment (AC2)

**Manuscript number: bg-2023-151**

**Response to reviewers**

Dear Simon Forster,

I would like to extend my sincere appreciation to the referees for their insightful comments and constructive feedback on this scientific manuscript. Your thoughtful inputs are invaluable and will significantly enhance the overall quality of the research.

Generally, the authors agree with most of the reviewers' comments and will use these inputs to better the quality of the manuscript

Below are the authors' comments on the review in general.

Review Summary

This paper provides new data regarding the benthic communities of oxygen minimum zones (in this case, the Northern Benguella Upwelling System), a generally under-studied habitat, and, as such is within the scope of BG. Unlike most macrofauna studies, the macro-nematodes (>0.5 mm) are included in the macrofauna analyses which provides a novel perspective.

The taxonomic discrimination of the general macrofauna was undertaken to 'major taxa' level whilst the macro-nematodes were identified to genus level. This allows for general comparisons with other studies to be undertaken and key patterns in macrofauna and macro-nematode distribution and hypoxia tolerances to be indicated and discussed.

The fact that the samples were sieved on 0.5 mm mesh precludes the entire nematofauna being considered (the majority of which fall within the 'meiofauna' category). This is, however, acknowledged and the term macro-nematodes is used correctly to distinguish these large species from the nematode community as a whole.

The text is, at times, confusing to the reader unfamiliar with the study and study site, and needs revision to enable an easier read. I have listed comments regarding this below along with any formatting errors/queries I have noticed.

The only significant scientific query I have is regarding the nematode feeding type categories applied to the various genera and, therefore, the conclusions drawn from these. This needs clarification and justification with references.

I think this paper should be published following revision and will complement the 'low-oxygen environments and deoxygenation in open and coastal marine waters' BG special issue.

Authors' response: Thank you

**Key Revisions and Clarifications Requested**

- The data analyses undertaken to justify the separation of the sites into the microxic, dysoxic and oxic zones (i.e. the SPSS) presents a trip-step in the results section. Consideration should be given to stating in the methods that these zone definitions were adopted (following Levin 2003) and removing section 3.2 from the results section. See comment for lines 130-134 and 162-174 below.

Authors' response: Thank you very much for your suggestion, the authors will heed the advice of the reviewer regarding lines 130-134.

- Clarification required regarding the Weiser feeding type allocation with justification of the allocation of Desmolaimus and Halanonchus to Type 2A. If these genera are subsequently considered Type 1B, the results and discussion will need revising accordingly.

Authors' response: Placing Desmolaimus and Halanonchus totype A was a mistake. The authors are really grateful for this and the results and discussions will be revised accordingly.

- Consideration should be given to including a fauna results summary table to include sample/site and community statistics (number of taxa, densities etc) that can be referred to when reading the narrative.

Authors' response: This is well noted and the faunal results summary table will be updated as suggested.

**Text and Presentation Comments**

Line 34 – Reference required for hypoxia/SDG statement.

Authors' response: The reference will be inserted.

More detail on the study location required for reader unfamiliar with the Namibian coast and the BUS:

Line 46 – Location details required. E.g. The Benguela upwelling system (BUS) is located off the West African coast extending from……….

Authors' response: The suggestion provided will be adopted by the authors.

Line 62 - Location details required. E.g. …….(namely, Walvis Bay, Namibia (the location of this study), California,…….)

Authors' response: The location details will be inserted as suggested.

Line 74 – Kunene river location details required. Reference to Figure 1 map with Kunene River labelled?

Authors' response: This will be addressed.

Lines 85/86 – reference to Figure 1 map required.

Authors' response: The reference to Figure 1 will be inserted.

Figure 1 – Needs more annotation for readers unfamiliar with the area. Small map needs country boundaries and names. Large map requires additional locations, namely, Luderitz and Kunene River, as these are mentioned in text.

Authors' response: The annotation on the map will be revised to help the reader understand the study area better.

Figure 1 – needs a colour scale bar or key for depth as this is mentioned in the site description (lines 85-89)

Authors' response: The authors will add a scale bar for the depth.

Line 94 – clarify whether 'high surface primary production' is mud surface or ocean surface. Obvious maybe, but avoid ambiguity.

Authors' response: The clarification will be made during the revision.

Line 101 – reference to Figure 1 required after at 90 nm.

Authors' response: Okay, the reference to figure 1 will be put after *..at 90 nm.*

Line 103 - reference to Figure 1 not required. Reference Table 1 instead

Authors' response: The authors agree, that referencing Table 1 rather than Figure 1 will make it easier for the reader to understand

Line 114-118 – please provide details of the level of taxonomy for non-nematode macrofauna taxa. Later, the term 'macrofauna taxa' is used without the reader knowing the definition of 'taxa'.

Authors' response: The details of the level of taxonomy for non-nematode macrofauna will be added during the revision period.

Line 119 – Need to state that feeding types were ascribed to genera following Weiser (1953) and insert reference in ref list.

Authors' response: The authors will state the ascribed feeding types of the nematodes and add the reference to the reference list

Line 130-144 – As above, it is not clear what level of taxonomic discrimination was used for statistical analyses.

Authors' response: As responded above, the authors will clarify the taxonomic classification of non-nematode macrofauna

Line 130-134 – This needs clarifying. Something along the lines of…..

Out of the seven sampled stations, only three had replicates. Therefore, direct statistical comparisons between sampling stations was not possible. To overcome this, the sampling stations were grouped into 'habitat types', based on the measured environmental variable that provided the most significant correlation with the various biotic indices. The communities within these key habitat types were then compared using indices and analyses outlined below.

An alternative approach that may be less complicated would be to state up front in this section that, following the approach of Levin (2003), the samples from microxic, dysoxic and oxic habitats will be compared in order to determine the effects of low DO. It could be further stated that this approach was justified by the application of SPSS that indicated DO was a key determinant associated with the microbenthic community structure. The text from lines 169-174 could be modified for this purpose.

Authors' response: The authors think that the alternative approach is an excellent suggestion and the authors will implement it during revision.

Line 134 – New paragraph starting at The data recorded….

Authors' response: The authors will put a new paragraph at the beginning of line 134 as suggested.

Line 142 – the various replicates were considered as a single samples. How was this done? If the replicate data were added together it would constitute 3 times the search effort as that for a single sample, therefore, the number of taxa will likely be elevated. Perhaps a more valid approach would be to randomly select a single replica for use in this analysis, rather than combine the three replicates?

Authors' response: Because the microxic and dysoxic stations had very few taxa and densities, random selection will exclude some taxa which may provide inaccurate results. The authors will consult and deliberate on the best approach to address the reviewer's concerns.

Line 147 – suggest term inverse relationship in preference to opposite trend.

Authors' response: This suggestion will be implemented during revision.

Line 147-150 – not easy to follow. Suggest

Total Organic Matter (%TOM) demonstrated an inverse relationship with depth, with higher organic matter values recorded in the shallower stations. For example, the shallowest stations 20020, 20002, and 23002 had the highest TOM of 38.6%±2.16, 27.5%±1.55 and 23.1%, respectively, whilst the lowest TOM values were found in the most offshore stations, stations 23070 and 26090, with 9.89%±1.55 and 4.4%, respectively (Table1).

Authors' response: The authors would like to thank the reviewer for his suggestion. This suggestion will be adopted during revision.

Line 151 - Similarly, dissolved oxygen values had lower values recorded in the shallower onshore station – According to Table 1, this is not correct; needs clarifying or removing. The OFC transect had

its highest DO at the shallowest site (33 m) and OWB had lowest DO at middle depth site (128 m), as described in the later text.

Authors' response: Thank you for pointing this out, the correction will be done during the review.

Line 158 – New paragraph after (Table 1).

Authors' response: The authors will put a new paragraph as suggested.

Lines 162-174 (3.2 Biotic Factors) – As discussed for Lines 130-134 above, it is suggested that the use of the approach of considering the sampling stations in terms of their DO zones (following Levin, 2003) be stated up front in the methods section. This section is somewhat of a trip-step in the narrative of the results.

Authors' response: The authors will address this when revising the manuscript and implementing the correction as suggested for Lines 130-134.

Line 171 – reference error. Should read (Levin, 2003).

Authors' response: Thank you for pointing this out, the correction will be done.

Line 162 – Section 3.2. As discussed above for Lines 130-134, it may be easier to follow if this narrative is moved to the data analyses methods section 2.4, to justify using the three DO habitat types.

Authors' response: Thank you for the suggestion. The author will move these lines and ideas to the data analysis methods.

Line 175 – 3.3 Macrobenthic assemblages. The term taxa is used with no definition (as mentioned for lines 114-118, above). Presumably, this is the taxonomic groups as listed on Figure 3, although the rarer are combined as others. Maybe list the taxa in a table?

Authors' response: The authors will list the taxa in a table for ease of reference.

Line 176-183 – This paragraph discusses the densities. The references to number of taxa are repeated in the subsequent paragraphs. Suggest the references to number of taxa are removed from this paragraph.

Authors' response: The reference to the numbers will be removed from the paragraph.

Line 215 – Insert reference to Figure 3, i.e. …… zones, respectively (see Figure 3).

Authors' response: Reference to Figure 3 will be inserted as suggested.

Line 216-219 – list of data is probably unnecessary as % abundance data already quoted. If a results summary table was included, this could be referred to instead.

Authors' response: Okay, the data will be removed and a result summary table will be created.

Line 222 – …..their abundance peaking in the Dysoxic station….. Is this relative abundances or total densities. If relative abundance, please state so and reference Figure 6 at the end of sentence.

Authors' response: The abundance in the sentence meant total densities. The sentence will further be clarified to enhance its readability and comprehensiveness.

Line 226 - …..recorded significant abundance……. If this is not a reference to statistical significance, please use alternative term, e.g. high abundance, to avoid ambiguity. Similarly, in line 232, please use low abundance unless statistically insignificant.

Authors' response: Point noted and the corrections will be done.

Line 230-234 – sentence long and complicated. Need clarifying and/or shortening. Suggest:

For the purposes of graphing the relative abundances, the genera Thoracostomopsis, Anticoma, Cephalanticoma, Trileptium, Mesacanthoides, Terschellingia, and Marylinnia were grouped as 'others' as they were recorded at low abundances (<4%); see Fig. 6.

Authors' response: Thank you for the suggestion this suggestion will be added to the revised copy.

Line 235-237 – For clarity, please include the feeding type code after the first mention of the feeding type, e.g. epistratum feeders (type 2A).

Authors' response: Thank you for the suggestion this will be added to the revised copy.

Line 239 – Clarification and justification is required regarding the allocation of feeding types. Whilst I appreciate that allocation of Weiser's feeding types is based on the morphology of the buccal cavities and is, therefore, partially subjective, most linhomoeid nematodes (with the exception of Terchellingia) are considered Type 1B (non-selective detritivore). I would classify Desmolaimus as Type 1B due to its unarmed (the cuticular 'arches' are unlikely to act as teeth) and relatively large buccal cavity. Similarly, I have always considered Halanonchus a Type 1B non-selective detritivore as, apart from buccal cuticular thickening, there is no dentition and has a large open mouth. Furthermore, these Genera tend to occur in muds. The results and discussion of feeding type guilds may need reviewing.

Authors' response: Thank you very much for pointing out this. The reviewer is right, Desmolaimus and Halanonchus are both 1B (non-selective deposit feeders. The authors will correct, review, and adjust the manuscript appropriately.

Line 256 – ….oxygen correlated significantly with….. Please state significance of correlation (p value?)

Authors' response: The p value will be added during revision.

Line 257 – Typographical error. Should read ………into three zones based on…..

Authors' response: Thank you for the correction, this will be corrected during revision.

Line 257 – for the benefit of the reader, it would be useful to remind of the three zones. Suggest inserting in parentheses, e.g. …….oxygen levels recorded (microxic zone (<0.1ml l-1); dysoxic zone (0.1-1.0 ml l-1); oxic zone (>1.0 ml l-1)).

Authors' response: Thank you for the suggestion this will be added to the revised copy.

Line 262-264 – It is essential to note……….their relative abundance. Does this mean polychaetes were recorded at their highest density at the microxic sites or they were the most abundant taxa in this zone? I assume the former. Needs clarification to avoid ambiguity.

Authors' response: This is well noted, the sentence will be clarified further.

Line 265 – Please define the core OMZ. OMZ spatial center? Area of lowest DO? Maybe in parentheses.

Authors' response: The core means the area of the lowest DO. The extra information will be added during review.

Line 273-275 – At the microxic zone……..in relative abundance. Meaning unclear. Does it mean, at the microxic sites, the polychaetes were the most abundant taxa in terms of total density, but the average relative abundance was lower than that of cumaceans? (How was the relative abundance of multiple sites calculated?)

Authors' response: Relative abundance was calculated for each core and then averaged. A hypothetical explanation can be for example, if one station had 3 cumacea and 1 polychaete, that means the relative abundance will be 75% and 25% respectively, then another station in the same oxygen zone has 5 polychaetes,  3 nematodes, and 2 cumacea. Polychaetes would have 50%, nematode 30%, and, 20% cumaceans. Hence the average relative abundance for cumacea will be about 47.5% and polychaetes will be 35%, despite polychaetes having 6 individuals while cumacea having only 5.

Line 276 - Were there only 4 specimens found at 23002, i.e. 3 cumaceans (75%) and 1 other taxa (25%)? If so, the 75% relative abundance, whilst correct, gives a misleading impression of the community and, when averaged with the other microxic station could be misinterpreted.

- This paragraph needs reviewing as it is not clear what point is being made and it read more like the results narrative than discussion.

Authors' response: This statement has proven to be problematic to both reviewers and I will therefore have to write it in an easier way for the reader to understand.

Lines 269, 274, 275 – The term cumacean should not be capitalised whereas Cumacea should. Please use terms consistently, either cumaceans or Cumacea wherever possible. Similarly, polychaetes or Polychaeta; nematodes or Nematoda etc. My preference would be polychaetes/nematodes etc when discussing the results, unless referring specifically to the phylum or class as a whole, in which case 'the Nematoda' or 'the Polychaeta' is good.

Authors' response: Thank you for sharing your preference, the terms polychaetes and nematodes will be used in the discussion area in the review.

Line 279 – please use the term macro-Nematoda or macro-nematodes, throughout. The Nematoda includes the meiofaunal size category whereas, this study, considers only those retained on a 500 μm mesh, i.e. the macrofauna size class.

Authors' response: Okay, this is well noted.

Line 286 – Should this read Apart from the increase in nematode size…

Authors' response: Thank you for the correction. The word nematode will be added to the review.

Line 289 – suggest ……coupled with a reduction in predation by larger fauna that are affected adversely by reduced oxygen concentrations…..

Authors' response: Thank you for the suggestion this will be added to the revised copy.

Line 296 – reference should read Gutierrez et al. (2008)

Authors' response: Thank you for the correction.

Line 301 – Once the DO levels to dusoxic levels…… This implies a temporal data set that documented an increase in DO. Suggest At dysoxic sites (DO 0.1 – 1 ml l1-), other taxa……

Authors' response: This suggestion is highly appreciated, the authors will adopt this suggestion.

Line 304 – as above. Suggest At the oxic sites, where DO levels were above 1.0 ml l-1, more taxa……

Authors' response: Thank you for the suggestion this will be added to the revised copy.

Line 308 – Lowest densities, diversity and species of macrofauna taxa or macro-nematodes or both? Please clarify.

Authors' response: This statement refers to general macrofauna taxa. Further edits will be made to minimize ambiguity.

Line 310 – As 301 and 304 above. Suggest At sites with DO above 1 ml l-1……..

- Does this refer to your study or Zettler's? Avoid ambiguity, suggest During the present study, at sites with….

Authors' response: The reference in this statement is regarding the current study. The statement will be edited to avoid ambiguity.

Line 314-316 – Most of ………. tolerance/intolerance to hypoxia. Meaning unclear. Suggest: Of these fauna, crustaceans were most abundant. This conforms to the observations of Soto et al. (2017) at oxic sites in an upwelling system in Chile. Conversely, Zettler et al. (2009) recorded amphipod species in low oxygen areas. These contradictory results indicate that, at least amongst the Amphipoda, tolerance/intolerance to hypoxia is species specific.

Authors' response: The authors appreciate this suggestion and they will be implemented during revision.

Line 357 – Suggest: Tolerance to hypoxia is indicated by both the presence and absence of taxa……..

Authors' response: Thank you for the suggestion this will be added into the revised copy.

Line 361 – insert Wieser reference

Authors' response: Wieser refrence will be added as suggested.

Line 362-363 – Need consistence of terminology for feeding types in text and Figure 7. Standard definitions are:

Authors' response: Thank you for the clarification, and the consistency of the terminology of feeding types.

Line 363 - Typographical errors – I think it should read ……(2A), with the former dominating the oxic zone.

Authors' response: Thank you for pointing this out, it's definitely a typographic error. It will be corrected when doing revision.

Line 367 – suggest These observations appear to be……

Authors' response: Thank you for the suggestion, this suggestion will be implemented.

Line 375-376 - The areas around the OMZ have also been observed to contain a high level of diatomaceous mud, which forms a significant component of epistratum feeders' diet, whose abundance was also high in the dysoxic areas.

Is it true that epistratum feeders (2A species) feed on diatomaceous mud? Need reference. 2A species feed on live diatoms whereas diatomaceous mud comprised dead, settled diatoms. Does the surface of the mud support epibenthic diatoms? The shallowest site was 33 m (oxic) whilst the rest of the oxic and dysoxic sites were in excess of 100 m where primary production would be very limited.

Authors' response: The authors will look into this statement and address it appropriately during revision.

Line 377 – Below the OMZ,…. Does this mean offshore from the OMZ, in deeper water?

Authors' response: Yes, below the OMZ means offshore from the OMZ, meaning the OMZ is no longer in contact with the benthic zone.

Line 385 – should this read Ostracoda and Bivalvia observed in limited numbers in the oxic zone…..? (not anoxic?)

Authors' response: From the manuscript, the word used is anoxic which is correct based on the results.

Line 380-391 (5.0 Conclusion) – the conclusion does not mention any thoughts on the macro-nematodes despite the title stating "a macro-nematode perspective".

Authors' response: Thank you for pointing this out. The authors will revamp the conclusion to include the macro-nematode perspective among other missing information.

---

## Author Comment (AC3)

**Manuscript number: bg-2023-151**

**Response to reviewers #2**

Dear reviewer,

I would like to extend my sincere appreciation to the referees for their insightful comments and constructive feedback on this scientific manuscript. Your thoughtful inputs are invaluable and will significantly enhance the overall quality of the research.

Generally, the authors agree with most of the reviewers' comments and will use these inputs to better the quality of the manuscript

Below are the authors' comments on the review in general.

**Referee 2**

Need to start with a sentence giving broader scientific context for your study.

Authors' response: The abstracts will be revised as requested.

Macrobenthic – use a small 'm', not capital 'M' (many similar mistakes in ms with Macrofauna,

Macro-nematoda, Dominance (!) etc…). I suggest do the same for oxic, microxic and dysoxic

Authors' response: Authors will revise the capitalization of these words as requested.

**Referee 2**

Instead of Macro-Nematoda, just use 'nematodes' once you have established that you are looking at macrofauna.

Authors' response: Authors will revise as suggested.

**Referee 2**

Line 18: 'were present' not 'recorded abundances' (correct throughout ms)

Line 21: 'were absent', not 'no abundance'

Authors' response: Authors will revise as suggested.

**Referee 2**

There is no concluding sentence at end of Abstract

Authors' response: Authors will include a concluding sentence at the end of the abstract.

**Referee 2**

Regarding the question on other studies that can be cited regarding the Namibian shelf,

Reverse order of 1st and 2nd sentence.

Authors' response: Authors will revise as suggested.

**Referee 2**

Line 45: need references at end of sentence.

Authors' response: Authors will add references as suggested.

**Referee 2**

Line 56: need more details on OMZ communities and their function.

Authors' response: Authors will add more details on OMZ communities and their functions.

**Referee 2**

Line 57-61 – too basic, delete. Just define macrofauna by short sentence in brackets when first

mentioning it.

Authors' response: Authors will revise as suggested.

**Referee 2**

Line 67 – need reference at end of this sentence

Authors' response: Authors will add references as suggested.

**Referee 2**

Nematodes are barely mentioned in the Introduction but they are main taxon of interest. Need info

on nematodes in general and in OMZ in particular.

Authors' response: Authors will add more details on nematodes and more focus will be given to
those from OMZ.

**Referee 2**

Line 73: specify where Walvis Bay is

Authors' response: Authors will revise as suggested.

**Referee 2**

Line 77: need reference at end of this sentence.

Authors' response: Authors will add references as suggested.

**Referee 2**

Line 78 – are there any other studies that can be cited in context of Namibian shelf?

Authors' response: The Authors will do further research and update the segment with more studies.

**Referee 2**

Line 82: you do not mention the focus on nematodes.

Authors' response: Authors will add more details on nematodes and more focus will be given to those from OMZ.

**Methods**

**Referee 2**

Line 95 – how low are the oxygen concentrations?

Authors' response: Based on Levin et al. (2009) the oxygen concentrations in was (<0.5 ml $L^{-1}$) in 55% of the total shelf whereas extreme anoxia (oxygen concentrations less than 1 µM) occurs over almost 900 km2

Table 1 – need to include number of replicates for macrofauna samples at each station

Authors' response: the number of replicates for macrofaunal samples at each station will be added during the revision period.

Line 114: why was 0.45 mm chosen for mesh size? Usually it is 500 or 300 microns. Not ideal for

comparing with other studies.

Authors' response: Typically, macrofaunal studies call for a sieve size of 0.5mm (500 microns). However, it is important to note that during the course of our study, the only available sieve was 0.45mm. As per the reviewer's comment, the usual sieve sizes are either 500 or 300 microns, and the 450-micron size falls between these two ranges. Furthermore, other studies have also used the 0.45 mm (450 microns) mesh size for macrofaunal studies like Li et al. (2018) https://doi.org/10.1016/j.ecolind.2017.11.003 and Zhang et al. (2022) https://doi.org/10.3390/d14121072.

Line 130 – this info needs to be in Table 1

Authors' response: the number of replicates for macrofaunal samples at each station will be added during the revision period.

Lines 130-134 – not easy to understand what you mean here. Just say what you did and why. How

were correlations conducted? What do you mean by 'highest correlation'? what variable was

chosen? Text in lines 169-174 needs to move from Results to Methods.

Authors' response: As for lines 130-134. Similar comments were made by the co-reviewer, the Authors will implement the suggestion by the first Authors to prevent a trip-step and reduce confusion, the Authors will state upfront the approach of Levin et al. (2003) was used to classify the stations into the various oxygen regimes. And then use the SPSS results to justify the classification. This will also result in the implementation of corrections pertaining to the comments on lines 141-144.

Line 138: need details of data treatment eg data transformation etc. Much too brief. What is anosim for? Simper?

Authors' response: The Authors will input more information as requested.

Line 141: list the predictor variables in text or table. What is 'BUS'??? What selection criterion did you use for your stepwise regression? R2? Aikaike?

Authors' response: Refer to the corrections made on lines 130-134

Line 142-144 – I don't understand what you mean. You matched biotic and abiotic data at scale of site for the correlation analyses?

Authors' response: Refer to the corrections made on lines 130-134

**Results**

Line 149: these are huge TOM values

Line 167: give R2 values and P values in text.

Authors' response: Authors will revise as suggested.

Before giving results of correlation analyses, describe the microbenthic assemblages first (section 3.3. before 3.2)

Authors' response: Thank you for the suggestions, the Authors will describe the macrobenthic assemblages first and revise as suggested.

Line 176: what statistical test was applied? P value? Of course they will be different because that's how you defined the groups to begin with.

Reorganise 3.3 to describe each group of stations – one paragraph per group.

Authors' response: Authors will revise as suggested.

Line 197: not 'taxa counts', just 'taxa'

Authors' response: Authors will revise as suggested.

Line 202: This sentence is much too vague. Dissimilar in what respect? Multivariate community structure? Statistically significant? P value? Pairwise comparisons? Show anosim table.

Authors' response: Dissimilar based on multivariate community analysis using Bray-Curtis analysis of dissimilarity.

Lines 216-219: delete, sentence confusing.

Authors' response: Authors will revise as suggested.

Line 234: need to do multivariate community structure analyses, as per the macrofauna taxa data.

Authors' response: Authors will do multivariate community structure analyses based on macrofauna data as suggested.

Line 235: why did you look at feeding guilds? There is nothing about this in the Methods or Introduction.

Authors' response: Prior mention and information regarding the feeding guilds will be included in the introduction and methods as suggested by the Authors.

**Discussion**

Line 257: clarify where the groupings come from. You use a scheme previously published?

Authors' response: The scheme was based on Levin (2003) and this clarification will be added to the revision.

Line 264: 'abundance', not 'quantities

Authors' response: Authors will revise as suggested.

Line 273-275: I don't understand. If a group has highest relative abundance then it has highest absolute abundance too.

Authors' response: Relative abundance was calculated for each core then averaged. A hypothetical explanation can be for example, if one station had 3 cumacea and 1 polychaete, that means the relative abundance will be 75% and 25% respectively, then another core in the same oxygen zone has 5 polychaetes, 3 nematodes, and 2 cumacea. Polychaetes would have 50%, nematode 30%, and, 20% cumaceans. Hence the average relative abundance for cumacea will be about 47.5% and polychaetes will be 35%, despite polychaetes having 6 individuals while cumacea having only 5.

Line 275-278: confusing sentence. Delete.

Authors' response: This is well noted and the Authors will heed this advice.

Line 286: Vanreusel et al. make no such statement as far as I know. Indicate where in paper they say this?

Authors''s response: Regarding the correction on line 286 inquiring about Vanreusel et al. (2010). The statement referenced can be found on page 3, paragraph 2:

> "Increased standing stock is not only explained by increased densities. Some studies [37,44] found that longer nematodes dominate in cold seep and hydrothermal sediments, compared to oxic neighboring sites. In [37], nematodes present in the hydrothermal vent are on average twice as large (800 µm long, 20 µm width), as those in the reference sediment (480 µm long, 15 µm width)."

Based on this study, macrofauna size was set at 0.45 mm meaning the 800 µm size is macrofaunal size.

Line 299: reference needed at end of this sentence Following sentence too vague. Why does patchiness 'call for more studies'?

Authors' response: To address patchiness and have a comprehensive representation of the study area, more studies is required. The study area hasn't been given the necessary research attention that it requires. This study found a high abundance of macro nematodes in one of the stations, to address if this phenomenon is characteristic of the study site or just a congregation to a food source will need more studies.

Line 302: 'low', not 'meager'

Authors' response: Authors will revise as suggested.

Line 303: families are not italicised.

Authors' response: Authors agree with the Authors, and shall revise as suggested.

Line 309: "1234 ind. m-2 recorded per core" makes no sense.

Line 317: delete brackets and text within. This whole paragraph just repeats same things mentioned before.

Authors' response: Authors will revise as suggested.

Line 329: you cite a review paper, you need to cite papers providing actual data.

Authors' response: Authors will revise as suggested.

Line 331: nematodes may be larger in some areas because the species are different. Unlikely to be because conditions give them ability to grow bigger

Authors' response: The nematodes were large in size, unfortunately, we didn't analyze the biomass due to the limited functionality of the microscope used.

Line 336: why would meiofaunal nematodes differ from macrofaunal nematodes?

Authors' response: Macrofauna and meiofauna are mainly separated based on size, as most meiofauna taxa are also found in the macrofauna component. As our study was mainly based on macrofauna, the presence of nematodes (whereby in most cases dominate the meiofauna component) were large in size and dominant in the dysoxic area.

Line 339: families are not italicised

Authors' response: Yes, the Authors agree with the reviewer, this was a typing mistake.

Line 342: need reference for this sentence

Authors' response: The Authors will add the reference as requested

Line 343: nematodes do not swim!

Authors response: Regarding the ability of nematodes to swim. The sentence was extracted from Moens et al., (2013) page 126, paragraph 1:

> "Nematodes can actively emerge into and swim in the water column (Jensen 1981). After suspension in the water column, some nematode species ( Theristus, Chromadorita, and Cobbia ) are able to actively choose and swim toward sediment spots where suitable food is available (Ullberg & Olafsson 2003). Large-bodied nematodes of the family Oncholaimidae rapidly colonize carcasses of fish and macrofauna, probably at least in part by active swimming (Lorenzen et al. 1987)."

Line 360: so what? Did you record Antcoma in your samples?

Authors' response: Yes, we recorded them only in the oxic zone. We will add more to the statement.

Line 361: Wieser

Authors' response: Correction well noted.

Line 367: re-write this paragraph. Outline your main findings, how they compare with previous findings, and interpret their meaning.

Authors' response: Authors will revise as suggested.

Line 381-2: delete this sentence

Authors' response: Authors will revise as suggested.

Line 387: did you identify species or just genera? Are you talking about nematodes?

Authors' response: The Authors identified the genera of nematodes and major taxa classification for other macrofauna. In the statement, the Authors are referring to both the macrofauna and the nematodes.

The word 'species' will be replaced with the word 'taxa'.

Line 389: ok, so overall your findings confirm what we know already or is there anything different?

Authors' response: The Authors will add more regarding the study's findings in line with other studies and the additional information the current study adds.

---

## Author Response (AR1)

**Influence of Oxygen Minimum Zone on Macrobenthic Community Structure in the Northern Benguela Upwelling System; A Macro-Nematode Perspective**

1.  **Reviewer 1**

| Serial No | Section | Reviewer 1 Corrections | Authors corrections |
|---|---|---|---|
| **1.0** | **Introduction** | Reference required for hypoxia/SDG statement | References added. |
| | | Line 46 - More detail on the study location required for reader unfamiliar with the Namibian coast and the BUS | Information added to read; The Benguela Upwelling System (BUS) is located off the southwest coast of Africa. It extends from Cape Frio in Angola to the southern tip of the continent in Cape Agulhas, South Africa. |
| | | Line 62 – Location details required | Location details added; The general trend observed in most OMZs in global oceans namely, Walvis Bay, Namibia (the location of this study), California, USA, and the Oman margin (off the Arabian Peninsula). |
| | | Line 74 – Kunene river location details required | Location details added; Sentence reads; In contrast the diversity increases significantly northwards off the Kunene River, which flows from the highlands of Angola, along the border with Namibia and into the Atlantic Ocean. |
| | | Lines 85 and 86 – Reference to figure 1 map required | Reference to (Figure 1) inserted at end of sentence. |
| | | Map needs annotation for readers unfamiliar with the area | New Map has been created and annotated. |

| | | | | |
|---|---|---|---|---|
| | | Map needs a color scale bar or key for depth | New Map has a color scale bar for depth with contours. |
| 2.1 | Study Area | Line 94 - Clarify whether 'high surface primary production' is mud surface or ocean surface | Clarified to specify ocean surface; Sentence reads; The benthic zone in the OMZ in Northern BUS is characterized by extensive areas of diatomaceous mud, which are associated with high primarily production at the ocean surface and low concentration of dissolved oxygen. |
| 2.2 | Sample Collection | Line 101 – Reference to Figure 1 required after…at 90nm | Reference to (Figure 1) inserted after 90nm. |
| | | Line 103 – Reference to Figure 1 not required. Reference Table 1 instead | Reference changed to (Table 1) instead. |
| | | Line 114 – 118; Provide details of the level of taxonomy for non-nematode macrofauna taxa | The taxonomical level of non-nematode fauna has been added |
| 2.3 | Laboratory Analysis | Line 119 - State that feeding types were ascribed to genera following Wieser (1953) and insert reference in ref list | Statement added to include this data; Now reads; The nematodes were then pin picked, fixed on permanent slides and identified to the genus level using the key from Platt & Warwick, 1988, and the feeding types were ascribed to those genera following the methodology of (Wieser, 1953) |
| 2.4 | Data Analysis | Line 130 – 134; It is not clear what level of taxonomic discrimination was used for statistical analysis. | Information on the taxonomic discrimination has been incorporated. |
| | | Clarify line 130 – 144 | Lines 130 – 144 have been altered and corrected. Paragraph now has 264 words. |

| | | | |
|---|---|---|---|
| | | New paragraph starts after line 134 | New paragraph starts after the end of this sentence. |
| | | Line 142 - How were the various replicates considered as single samples? | This has been tackled in the revamped data analysis section. |
| | | Line 147 – Suggest term inverse relationship in preference to opposite trend | Change implemented, now reads; Total Organic matter (%TOM) demonstrated an inverse relationship with depth, with higher organic matter values recorded in the shallower stations. |
| 3.1 | **Abiotic Variables** | Line 147 – 150 – Not easy to follow | Sentence structure altered for clarity; Collective word count is 61. |
| | | Line 151 is not correct based on Table 1, needs clarifying or removing | Sentence not in line with Table 1 deleted. |
| | | Line 158 – New paragraph after (Table 1) | New paragraph now starts after the reference to (Table 1) |
| | | Line 171 – Reference error, correct | Reference corrected; now reads (Levin, 2003) |
| 3.2 | **Biotic Factors** | Line 162 is not clear | Paragraph from line 162 – 174 has been deleted. Information has been incorporated into Section 2.4; Data analysis. |
| 3.3 | **Macrobenthic Appendages** | Line 175 – 3.3 Macrobenthic assemblages. List the taxa in a table? | Taxa list added to supplementary material |
| | | Line 176 – 183 – References to number of taxa removed from paragraph | All references to number of taxa have been deleted. Paragraph now focuses solely on densities; has 83 words. |
| | | Line 215; Insert reference to Figure 3 | Reference added, reads (See Figure 3) |

| 3.4 | Macro-nematodes density and diversity | Line 216 – 219; List of data is unnecessary as % abundance data has already been quoted | List of unnecessary data deleted. Previous lines 216 to 219 are no longer present. |
|---|---|---|---|
| | | Line 222 – Clarify what relative abundances you are talking about. State and reference Figure 6 at end of sentence. | Statement now reads, "exhibited their highest densities in dysoxic stations" for clarity. |
| | | Line 226 – recorded significant abundance is ambiguous, use alternative term | Term 'significant abundance' has been altered to 'high abundance' to clearly refer to the total densities. |
| | | Line 232 – use term low significance | 'Insignificant abundance' replaced with the term 'low significance' |
| | | Lines 230 – 234; Sentence long and complicated. Needs clarifying and/or shortening. | Sentence clarified; Now reads; For the purposes of graphing the relative abundance, *Thoracostomopsis, Anticoma, Cephalanticoma, Trileptium, Mesacanthoides, Terschellingia,* and *Marylinnia* were grouped as 'others' as they recorded low abundances (<4%). |
| | | Lines 235 – 237; For clarity include the feeding type code after the first mention of the feeding type | Feeding type codes have now been incorporated; Statement altered to; Epistratum feeders, classified as Type 2A, dominated the dysoxic zones with a proportion of 62%. They were followed by predators/omnivores, Type 2B, making up 28%. Lastly, selective deposit feeders, classified as Type 1A, constituted 10% of the population. |

| | | Line 239; Clarify the feeding types, Desmolaimus and Halanonchus are incorrectly grouped | Corrected accordingly all through the manuscript |
|---|---|---|---|
| **4.0** | **Discussion** | Line 256; What had significant correlation with oxygen? | The sentence was structured based on the direction given by the co-reviewer. |
| | | Line 257; Correct typographical error | Error corrected. Classification into zones based on oxygen levels now captured in new sentence. |
| | | Line 237; For readers benefit, mention the three zones, preferably by inserting parentheses | The three zones of classification are reiterated, with their corresponding values in parentheses; The statement now reads as follows. We adapted Levin's grouping system (Levin, 2003), classifying the different stations into zones based on the oxygen levels recorded (microxic zone ($<0.1$ ml l$^{-1}$); dysoxic zone ($0.1$-$1.0$ ml l$^{-1}$); oxic zone ($>1.0$ ml l$^{-1}$)). |
| | | Line 262; Clarify what you mean by relative abundance, sentence is confusing | Some information has been added after line 262 to supplement this ambiguity. Relative abundance is described in line 263, reading; Here, relative abundance refers to the proportion of polychaetes to the total number of organisms in the same area. Therefore, even though polychaetes were numerically abundant, the diversity of other taxa present reduced their share of |

| | | | | the total population, hence the low relative abundance. |
|---|---|---|---|---|
| | | | Line 265; Please define core OMZ – OMZ special center? Area of lowest DO? | To specify core OMZ to the area of lowest dissolved oxygen, sentence now reads; The presence of cumaceans in high abundance in the core OMZ (Area of lowest DO) has been reported by Zettler et al., (2013) and Eisenbarth & Zettler (2016) |
| | | | Line 273 – 275; Sentences unclear, how was the relative abundance of multiple sites calculated? | Entire paragraph has been amended to include information on how relative abundance is calculated. Paragraph has 143 words. |
| | | | Line 276; Paragraph needs review as the point is not clear, reads more like the results narrative than discussion | Entire paragraph revamped. Has 143 words. |
| | | | Line 269, 274, 275; Establish consistency in taxonomic names; Preference is polychaetes/nematodes when discussing the results unless referring specifically to the phylum or class, in which case 'the Nematoda' or 'the Polychaeta' is good | Taxonomic names now have a consistency in the entire manuscript |
| | | | Line 279; Use the term macro-nematoda or macro-nematodes throughout. | Changed from 'Nematoda' to macro-nematoda. |
| | | | Line 286 – Change sentence structure | Now reads; Apart from the increase in nematode size, OMZs also tend to enhance the regional dominance of tolerant organisms such as nematodes with high biomass |

| | | | | recorded in response to organic matter inputs. |
|---|---|---|---|---|
| | | Line 289; | | 'Coupled with release from predation' replaced with 'coupled with a reduction in predation' |
| | | Line 296 – Reference should read Gutierrez et al. (2008) | | Reference now reads Gutiérrez et al. (2008) |
| | | Line 301 suggests a temporal data set that documented an increase in DO. Suggest; At dysoxic sites (DO 0.1 – 1 ml-1), other taxa | | 'Once the DO levels rise to dysoxic levels' replaced with 'At dysoxic sites (DO 0.1 – 1 ml-1) to correct this. |
| | | Line 304; Follow as suggested in line 301 | | 'When the DO levels increased to above 1.0 ml l$^{-1}$ replaced with 'At dysoxic sites, where DO levels were above 1.0 ml l$^{-1}$). |
| | | Line 308; Lowest densities, diversity, and species of macrofauna taxa or macro-nematodes or both? Clarify. | | Reads; In the core (microxic) area, the macrofauna taxa showed the lowest density and diversity to specify that it is the general macrofauna taxa that is the subject of this information. |
| | | Line 310 – Follow as with lines 301 and 304. Does this refer to your study or Zettler's? Avoid ambiguity | | Reference to this study mentioned in the sentence; In our study, we also observed an increase in the number of taxa recorded in sites with DO levels above 1 ml l$^{-1}$ to avoid ambiguity. |
| | | Line 314 – 316; Meaning unclear, correct. | | Unclear statement corrected as follows; Of this fauna, crustaceans were the most abundant. This conforms to the observations of Soto et al. (2017) at oxic sites in an upwelling system in Chile. |

| | | | Conversely, Zettler et al. (2009) recorded amphipod species in low oxygen areas. These contradictory results indicate that, at least amongst the Amphipoda, tolerance/intolerance to hypoxia is species specific. |
|---|---|---|---|
| | | Line 357; Change sentence structure | Structure changed to; Tolerance to hypoxia is indicated by both the presence and absence of taxa. |
| | | Line 361; Insert Wieser reference | Reference added; Wieser's feeding types, as outlined in his study (Wieser, 1953) |
| | | Line 362 -363; Need consistence of terminology for feeding types in text and figure 7. | Consistency in feeding type terminology and code has been implemented throughout the manuscript. |
| | | Line 363; Correct typographical errors | Corrected, Epistratum feeders now classified as type 2A, not type 1B |
| | | Line 367; Alter sentence structure | Sentence altered; Now reads; These observations appear to be exceptions to the general rule that non-selective deposit feeders dominate substrates with a high abundance of organic matter. |
| | | Line 375 – 376; Is it true that epistratum feeders (2A species) feed on diatomaceous mud? Needs reference. | Authors have changed the statement based on the feeding mode correction. Also the statement has been changed to diatoms and not diatomaceous mud. |

| | | Line 377 – Below the OMZ…Does this mean offshore from the OMZ, in deeper water? | To clarify 'below the OMZ' means offshore from OMZ, sentence has been clarified to read; In regions offshore from the OMZ, where the OMZ is no longer in contact with the benthic zone, the production of diatoms is reduced. |
|---|---|---|---|
| **5.0** | **Conclusion** | Line 385 – Should this read Ostracoda and Bivalvia observed in limited number in the oxic zone…? (not anoxic?) | The word 'oxic' has been replaced with 'anoxic' |
| | | Line 380 – 391; The conclusion does not mention any thoughts on the macro nematodes despite the title stating "a macro-nematode perspective" | The conclusion has been restructured |

**Reviewer 2**

| Serial No | Section | Reviewer 2 Suggestions | Authors corrections |
|---|---|---|---|
| | Abstract | Need to start with a sentence giving broader scientific context of your study | Statement providing broader scientific context of the study has been inserted at the start of the Abstract. Sentence has 42 words. |
| | | Macrobenthic – Use a small 'm', not capital 'M' (many similar mistakes in MS with Macrofauna, Macro-nematoda, Dominance. | Capitalization of letters relating to macrofauna, macro-nematoda, dominance, anoxic, dysoxic and oxic has been addressed, apart from first mentions and those in the start of sentences. |
| | | Instead of Macro-Nematoda, just use 'nematodes' once you have established that you are looking at macrofauna | Change has been implemented, macro-nematoda is used fewer times after first mention. |
| | | Line 18 – not 'recorded abundances' throughout MS | The term 'Recorded abundances' has been removed and replaced with the term 'were present' This change has been implemented all through the manuscript |
| | | Line 21 - 'no abundance' is incorrect | 'No abundance' has been replaced with 'absent' |
| | | There is no concluding sentence at end of the Abstract. | Conclusive sentence added to the terminal end of the abstract. Reads; In conclusion, this study provides an overview on the distribution, diversity, and response to varying oxygen conditions of macrobenthic communities and their importance in marine ecosystems. |
| 1.0 | Introduction | Reverse order of 1st and 2nd sentence. | Sentence order reversed |

| | | Line 56 needs more details on OMZ communities and their function. | More details on OMZ communities have been added. The supplementary statements have 74 words. |
|---|---|---|---|
| | | Line 57 – 61 are too basic. Define macrofauna by short sentence when first mentioning it. | Lines 57 – 61 have been deleted and in their stead, the term macrofauna is explained by a short statement in parentheses in line 65. |
| | | Line 67 needs a reference. | Reference added |
| | | Nematodes are barely mentioned in the introduction but are the main taxon of interest. More information on nematodes is needed, and OMZs. | More information on nematodes and OMZs and their communities has been added to the introduction section. |
| | | Specify the location of Walvis Bay | The location of Walvis Bay had been supplemented. Walvis Bay, a city located on the Western coast of Namibia is the new descriptive addition. |
| | | Line 77 needs a referenced | Reference added |
| | | Are there any other studies that can be cited in context of the Namibian shelf | Reference added |
| | | There is no focus on nematodes in line 82 | More focus has been put on nematodes and OMZs, new supplementary statement has 78 words. |
| | | How low are the oxygen concentrations in line 95? | Information on Oxygen concentrations has been added, referenced to Levin et al. (2009) |
| | | Table 1 needs number of replicates for macrofaunal samples at each station | Information on replication added |
| | | Why was 0.45 mm chosen as the mesh size? Usually, it is 500 or 300 microns. | Information on reason for the use of 0.45 mm sieves and the validity of |

| | | | |
|---|---|---|---|
| | | Not ideal to use 0.45 when comparing with other studies. | the results has been added in parentheses in line 127 |
| 2.1 | **Data analysis** | Information on line 130 needs to be in Table 1 | Information on replication added |
| | | Lines 130 – 134 are not easy to understand. | The entire paragraph has been altered to cater for this confusion. Paragraph has 264 words. |
| 2.4 | **Data analysis** | Line 138 needs details of data treatment, it is much too brief. | No data treatment was used. |
| | | List the predictor variables in a text or table. What is BUS? and what selection criterion did you use? | Predictor values are added on the supplementary file. 'BUS' replaced with 'Benguela Upwelling System' |
| | | Lines 142 – 144 are confusing; did you match biotic and abiotic data at scale of site for the correlation analysis? | Lines 142 – 144 have been corrected as part of previous suggestion and are no longer confusing. |
| 3.1 | **Abiotic variables** | These are huge TOM values is line 149 | Yes, these were the observed TOM values from the study site |
| 3.2 | **Biotic factors** | Move lines 169 – 174 from results to methods | Lines 169 – 174 have been moved to data analysis methods under the second paragraph of the section. |
| | | Add R2 values and P values in your text in line 167 | Added in line 224 |
| 3.3 | **Macrobenthic assemblages** | Before giving results of correlation analysis, describe the macrobenthic assemblages first (section 3.3 before 3.2) | Entire section 3.2 has been deleted to avoid the trip up in the flow of information. (Section 3.3 Macrobenthic assemblages) is now Section 3.2 |
| | | Reorganize section 3.3 to describe each group of stations – one paragraph per group | Section 3. 3 reorganized with each paragraph focusing on the different stations namely; microxic, oxic and dysoxic. |

| | | | |
|---|---|---|---|
| | | Not 'taxa counts', just 'taxa' in line 197 | 'Taxa counts' has been replaced with 'taxa' |
| | | Line 202 is vague, needs clarifying | Information added to cater for vagueness; the sentence now reads; All the oxygen zones were dissimilar to one another based on multivariate community analysis using Bray-Curtis's analysis of dissimilarity. |
| 3.4 | **Macro-nematodes density and diversity** | Lines 216 – 219 are confusing. Delete | Lines 216 – 219 have been deleted from the manuscript |
| | | Line 234 shows need to do multivariate community structure analyses as per the macrofauna taxa data | For clarity purposes, the Multivariate analysis results were not included to maintain focus on dissolved oxygen (DO). However, the authors are open to incorporating the multivariate analysis if the reviewer suggests it. |
| | | There is nothing about feeding guilds in the methods or introduction | Feeding guilds were added to the methods. |
| 4.0 | **Discussion** | Clarify where the groupings come from in line 257, did you use a previously published scheme? | Reference to Levin's grouping system based on the levels of oxygen recorded has been added to line 257 to point out where the groupings came from; Sentence reads; "We adapted Levin's grouping system (Levin, 2003), classifying the different stations into three zones" |
| | | Line 264 - Quantities is not the correct term, use abundance. | 'Quantities' has been replaced with 'abundance' |

| | | | |
|---|---|---|---|
| | | Information in lines 273 – 275 is not understood well. | New information has been added to clarify the subject in lines 273 - 275 |
| | | Lines 275 – 278 are confusing. | Sentence structure and new information have been added to avoid this confusion. |
| | | Vanreusel et al. makes no such statement, indicate where in the paper they say this? | Vanreussel does make this statement. Reference to page 3, paragraph 2: "Increased standing stock is not only explained by increased densities. Some studies [37,44] found that longer nematodes dominate in cold seep and hydrothermal sediments, compared to oxic neighboring sites. In [37], nematodes present in the hydrothermal vent are on average twice as large (800 μm long, 20 μm width), as those in the reference sediment (480 μm long, 15 μm width)." |
| | | Insert reference at the end of line 299 | Reference added. |
| | | Sentence following line 299 is vague. Why does patchiness call for more study? | 'It is not clearly understood then whether the high abundance of macro nematodes in one of the stations is characteristic of the study site or just congregation to a food source' has been added to precede line 299, hence justifying the reason for more study. |
| | | 'meager' is not the correct tern | 'meager' replaced with 'low' |
| | | Families are not italicized | Italics removed from the family name. |

| | | | |
|---|---|---|---|
| | | Line 309's "1234 ind. m-2 per core" makes no sense. | To correct this, sentence has been changed to; "Each square meter of core area contained 1243 individuals." |
| | | Line 317; delete brackets and text within. The whole paragraph is repeating whats already been mentioned. | Entire paragraph has been deleted from the manuscript. |
| | | Line 329 cites a review paper, need to cite papers providing actual data. | Reference added |
| | | Line 331 is unlikely, nematodes may be larger because of the species, not because of the conditions. | Statement 'ability to grow to large sizes' has been removed. |
| | | Why would meiofaunal nematodes differ from macrofaunal nematodes? | Macrofauna and meiofauna are mainly separated based on size, as most meiofauna taxa are also found in the macrofauna component. As our study was mainly based on macrofauna, the presence of nematodes (whereby in most cases dominate the meiofauna component) were large and dominant in the dysoxic area. |
| | | Line 339 - Families are not italicized | Italics have been removed. |
| | | Line 342 needs reference | Reference added |
| | | Line 343 is incorrect, nematodes do not swim! | Regarding the ability of nematodes to swim. The sentence was extracted from Moens et al., (2013) page 126, paragraph 1: "Nematodes can actively emerge into and swim in the water column (Jensen 1981). After suspension in the water column, some nematode |

| | | | species ( Theristus, Chromadorita, and Cobbia ) are able to actively choose and swim toward sediment spots where suitable food is available (Ullberg & Olafsson 2003). Large-bodied nematodes of the family Oncholaimidae rapidly colonize carcasses of fish and macrofauna, probably at least in part by active swimming (Lorenzen et al. 1987)." |
|---|---|---|---|
| | | Line 360; Provide recordings of Anticoma in your samples. | Anticoma was not recorded in the study. |
| | | Typographical error in name Weiser in line 361 | 'Weiser' changed to 'Wieser' |
| | | Rewrite the entire of paragraph beginning at line 367 | Paragraph has been rewritten to clarify information |
| | | Delete line 381 | Line 381 is no longer part of the manuscript. |
| **5.0** | **Conclusion** | In line 387, state whether you identified species or just genera, clarify whether you are talking of nematodes. | 'species' has been changed to 'taxa' for specificity. |
| | | Are there any differences in your findings or are they in confirmation of data already known? | The differences and similarities have been mentioned in the discussion. |

---

## Author Response (AR2)

**Influence of Oxygen Minimum Zone on Macrobenthic Community Structure in the Northern Benguela Upwelling System; A Macro-Nematode Perspective**

**Response to the Editor's comments**

Thank you for the corrections. Based on the line numbers, I deduced that these corrections were made from the Author's track changes (ATC) file. Some of the corrections were already made in the clean copy but I will still mention them in this document, however, the line numbering will be based on the last clear manuscript (version 3).

Please note that after I stopped the tracking changes and had the clean manuscript, I made some small changes to the manuscript like commas, and slight changes in words to enhance the clarity of the manuscript, thus the final manuscript version might have more corrections compared to the ATC file

| Serial No | Section | Reviewer 1 Corrections | Authors corrections |
|---|---|---|---|
| | **Abstract** | Do not capitalize Macrobenthic on line 20 or Diversity on line 25 (both are mid sentence) | Both 'macrobenthic' and 'diversity' have been corrected. Refer to lines 20 and 25 respectively. |
| **1.0** | **Introduction** | Line 96 – macrobenthis that are 2 to 0.5 mm are not large enough to be seen with the naked eye! That definition is for megafauna. | This statement has been corrected to read "benthic organisms that are typically retained in a 0.5 mm sieve but pass through a 2.00 mm sieve" (Line 73-74) |
| | | Line 99 – The citation Gibson & Atkinson, 2003 should be Levin, Lisa A. 2003. (Gibson and Atkinson are the OMBAR editors for all the volumes in all the years!). | The citation has been corrected and the in-text reference '(Levin, 2003) was adopted as this has been in use throughout the manuscript. (Line-63 & 76) |

|  |  |  | Line 113 – CHANGE Spinoid, Dorvilleid, and Lumbrinerid to the family names Spionidae, Dorvilleidae and Lumbrineridae. | Corrections have been made to the polychaete families (Line 80) |
|---|---|---|---|---|
|  |  |  | Line 123 change reducing latitude to lower latitude | 'Reducing' has been changed with 'lower'. Line 95 |
| 2.2 | **Sample Collection** |  | Line 154 Please indicate in this sentence if these are distances from shore – or if not, distances from what? The sampling stations were located at 02 nm, 20 nm, 40 nm, or 70 nm at each transect, with the 26º S transect hosting only one station at 90 nm (Figure 1). | The word 'from the shore' has been added in **Line 123**. |
| 2.3 | **Laboratory Analysis** |  | Line 174 add 'for macrofauna' after (300-500 microns). | The correction has been made and the statement reads "This size, although slightly smaller, falls within the range of commonly used sieve sizes (300 to 500 microns) for macrofaunal research." Refer to Line 141. |
|  |  |  | Line 193. Add 'They are before further subdivided… | 'they are' has been added as requested. See Line 154. |
|  |  |  | Line 197 – do not capitalize Selective | The letter 's' has been corrected as requested. (Line 158). |
|  |  |  | Line 206 remove the second 'then' | The first 'then' has been removed (Line 167) |
| 2.4 | **Data analysis** |  | Give the log scale for H' e.g., H'log10 or H'loge | Further information on the calculation of the diversity indices |

| | | Explain what metric dominance is … rank 1 dominance? (or J'?) | has been added in section '2.4 Data Analysis'. Refer to lines 190-191. |
|---|---|---|---|
| 3.1 | **Abiotic Factors** | Line 278, 282 Do not use zeros for degree sign | The degree symbol has been corrected throughout the manuscript. See lines 207, 209, and 211. |
| 3.2 | **Macrobenthic Appendages** | Line 291 Replace contrastingly with in contrast | 'Contrastingly' has been replaced with 'in contrast'. Refer to line 235. |
| | | Line 330 and 544 replace taxa with taxon | Taxa has been replaced by 'taxon' in Line 225. |
| | | Line 276 Superscript the -1 in ml l-1 Line 291 and 293 and 331, 335 etc. – superscript the -2 on m-2. | Superscripts have been inserted in the instances mentioned. |
| 3.3 | **Macro-nematodes density and diversity** | Line 467 do not capitalize Diversity | Diversity has been written in small letters. (Line 292). |
| 4.0 | **Discussion** | Line 500 Change groupings to grouping | The cleaner version was reading 'grouping', so no further action was taken. (Line 304) |
| | | Line 509 to 512. This is redundant to the previous sentence. Rephrase to say It is essential to note that numerically Polychaeta was the most abundant in this oxygen zone, RELATIVE TO OTHER ZONES, but the presence of MANY INDIVIDUALS OF other taxa in the MICROXIC stations reduced | The suggestions of the editor were adopted, however, the term 'relative to other zones' was not included because it meant that Polychaeta was numerically more abundant than in the microxic zone compared to other zones which was not the case. |

| | | | |
|---|---|---|---|
| | | their PROPORTIONAL REPRESENTATION. . OMIT _ the text - Here, relative abundance refers to the proportion of 509 polychaetes to the total number of organisms in the same area. Therefore, even though polychaetes 510 were numerically abundant, the diversity of other taxa present reduced their share of the total 511 population, hence the low relative abundance. | The statement now reads "It is essential to note that numerically Polychaeta was the most abundant taxon in this oxygen zone, but the presence of many individuals of other taxa in microxic stations reduced their proportional representation." Lines 312-314. |
| | | Line 513 – indicate if the OMZ core is comparable to the microxic zone in your study | The OMZ core's comparison with our study has been included. Refer to Line 314 |
| | | Line 520 – change quantities to concentrations | 'Quantities' has been changed to 'concentrations'. Refer to line 322 |
| | | Line 534 and 539– remove parentheses from around station number | Parentheses have been removed. Refer to Lines 325-343. |
| | | Line 540 use the same nomenclature for individuals per square metre, (ind. m-2) throughout the paper | The nomenclature has been harmonized to ind. $m^{-2}$. Refer to line 342. |
| | | Line 581 Change alluded to suggested | The term 'suggested' replaced 'change'. Line 364 |
| | | Line 629 – Hyphenate species specific | The hyphen had been inserted in the clear copy. No further action was taken. Refer to Line 396. |
| | | Line 688 insert a semicolon after taxa and before most genera | The semicolon was already inserted in the clear copy. No |

| | | | |
|---|---|---|---|
| | | | further action was taken. Refer to Line 429. |
| | | Line 718-9 Italicize the three genera of bacteria. | The genera have been italicized. Refer to line 443-445. |
| | | Line 732 insert space after (2B) | Space has been inserted after (2B) |
| **5.0** | **Conclusion** | Line 776 is there a missing word? What does 'co' represent? Counterparts? | Yes, the 'co' was supposed to be counterparts. Line 478 |
| | **Table & Figures** | Line 813 Table 1. change heading below Station from Replicate to No. Replicates | 'No. Replicates has been added on the 2nd column. |
| | | Figure 2 caption. Explain what the error bars are – +1 SD? +1 SE? | Error bars has been explained at the end of the caption. |
| | | Figure 3 and 6 caption. Do not capitalize relative. | The 'r' in relative has been written in lowercase. |
| | | Figure 5 and Figure 8 caption. Give the log base of H'. Explain which dominance metric is used here – is this J'? Rank 1 Dominance? Explain what the error bars are. + 1 SD? + 1SE? | H' diversity, dominance, and error bars have been explained in these figures |
| | | Fig. 6 legend. Are you able to italicize the genera? | Yes, The genera have been italicized. |
| | | Figure 7 Add (1953) after Wieser | (1953) has been added to the figure caption. |
| | | Acknowledgements. Line 2 remove extra (, | The extra ( has been removed. |

|  |  |  |  |
| --- | --- | --- | --- |
|  |  | Consider thanking the (anonymous) reviewers for input? | The reviewers' inputs have been acknowledged. |
|  | **References** | Gibson, R. N., & Atkinson, R. J. A. (2003). Oxygen minimum zone benthos: adaptation and community response to hypoxia. Oceanogr. Marine Biol. Annu. Rev, 41, 1–45. Should be cited as: Levin, Lisa A. (2003). Oxygen minimum zone benthos: adaptation and community response to hypoxia. Oceanogr. Marine Biol. Annu. Rev, 41, 1–45. | The reference 'Levin, Lisa A. (2003). Oxygen minimum zone benthos: adaptation and community response to hypoxia. Oceanogr. Marine Biol. Annu. Rev, 41, 1–45' was already on the reference list as 'Levin Lisa. (2003). Oxygen minimum zone benthos: Adaptation and community response to hypoxia. In Gibson R. N & Atkinson R.J.A (Eds.), Oceanography and Marine Biology: An Annual Review (Vol. 41, pp. 1–45). CRC Press.' |